# *ERECTA* genes and their ligands regulate shoot and inflorescence architecture in maize

Xiao Liu [1,9], Jinbiao Wang [1,9], Jipeng Li [1], Lu Kang[2], Mengyan Wang [1], Zhaoyu Huang [1], Jarrett Man[3], Xuxu Huang[1], Zhiming Zhang[4], Fang Yang [2], Madelaine Bartlett [5], Liuji Wu [6], Zhaobin Dong [7], David Jackson [8]✉ & Fang Xu [1]✉

In maize, several yield-related traits are associated with meristem activity, regulated by CLAVATA3/EMBRYO SURROUNDING REGION-related (CLE) peptide signals perceived by CLAVATA(CLV) receptors in the CLAVATA-WUSCHEL (CLV-WUS) pathway. However, additional signaling pathways in maize meristem development remain poorly understood. Here, we identify three receptor-like kinases, ZmERECTA1 (ZmER1), ZmER2 and ZmER1-like (ZmERL), and their ligands, EPIDERMAL PATTERNING FACTOR-like (ZmEPFL), as critical regulators of meristem activity, plant architecture, and ear development. We demonstrate that ZmER receptors act redundantly, with ZmER1 playing a primary role. *Zmer1* mutants have compact architecture, enlarged inflorescence meristems (IMs), and increased kernel row numbers (KRNs), while higher-order *Zmer* mutants display exacerbated phenotypes. We further reveal that ZmER1 specifically binds to five EPFL peptides, which act redundantly in ear development regulation. Furthermore, we find that *ZmWUS1* is upregulated in *Zmer* mutants and mutation in *Zmwus1* partially suppress the enlarged IM of *Zmer1* mutants. We also generate weak *Zmer1* alleles with enhanced yield traits, including reduced leaf angles and increased KRN. These findings offer valuable insights into ER-EPFL signaling in maize meristem development and provide promising genetic targets for breeding high-yield maize varieties through optimized plant and ear architecture.

Maize (*Zea mays*) is a worldwide cereal crop, and is important to meet increasing demands for food, feed and biofuel consumption. Maize yield is a complex trait depending on many factors, including plant and inflorescence architecture[1–5]. Shoot architecture is derived from the stem-cell-containing shoot apical meristem (SAM), which maintains self-renewal as well as incorporates cells into new leaf primordia and axillary shoot meristems or stem tissues[6,7]. The development and regulation of SAM activity play an important role in shaping multiple

[1]The Key Laboratory of Plant Development and Environmental Adaptation Biology, Ministry of Education, School of Life Sciences, Shandong University, Qingdao, China. [2]School of Agriculture and Biotechnology, Sun Yat-Sen University, Shenzhen, China. [3]Biology Department, University of Massachusetts, Amherst, MA, USA. [4]State Key Laboratory of Crop Biology, College of Life Sciences, Shandong Agricultural University, Taian, China. [5]Sainsbury Laboratory, Cambridge University, 47 Bateman Street, Cambridge CB2 1LR, UK. [6]State Key Laboratory of High-Efficiency Production of Wheat-Maize Double Cropping, College of Agronomy, Henan Agricultural University, Zhengzhou, China. [7]State Key Laboratory of Maize Bio-breeding, National Maize Improvement Center, Frontiers Science Center for Molecular Design Breeding, China Agricultural University, Beijing, China. [8]Cold Spring Harbor Laboratory, Cold Spring Harbor, New York, NY, USA. [9]These authors contributed equally: Xiao Liu, Jinbiao Wang. ✉e-mail: jacksond@cshl.edu; fxu@sdu.edu.cn

key plant architecture traits[6–8]. The seed-bearing ear develops from inflorescence meristems (IMs), which give rise to a stereotypical series of spikelet and floral meristems that form kernels after fertilization[9]. Key ear traits including kernel row number and kernel number per row are highly dependent on IM activity[10,11].

At the center of both SAM and IM maintenance is the highly-conserved CLAVATA (CLV)-WUSCHEL (WUS) regulatory feedback pathway[12,13]. In maize, the leucine-rich repeat (LRR) kinase THICK-TASSEL DWARF 1 (TD1) and LRR proteins FASCIATED EAR 2 (FEA2) and FEA3 function as CLV-related receptors, which together perceive various CLE peptide signals, including ZmCLE7, ZmCLE1E5, and ZmFON2-LIKE CLE PROTEIN1 (ZmFCP1)[14,15]. Additionally, the pseudokinase CORYNE (ZmCRN) and G protein α subunit COMPACT PLANT 2 (CT2) serve as downstream effectors of FEA2 to transmit distinct ligand signals[14,16]. Mutations in these CLV-related genes increase meristem size, creating additional space for producing more kernel rows and displaying a fasciated ear phenotype. Fine-tuning meristem activity through genetic manipulation of CLV-related genes, either by mutations in protein coding or cis-regulatory regions, presents new possibilities for improving yield traits[1,4,14,17].

In addition to the CLV signaling receptors and ligands, the ERECTA (ER) receptor family, which mediates EPIDERMAL PATTERNING FACTOR-Like (EPFL) peptide signaling, is another prominent pathway for the regulation of meristem size and inflorescence architecture in the model plant *Arabidopsis thaliana* (arabidopsis)[18–20]. The arabidopsis ER family (ERf) genes include ERECTA (ER), ERECTA-LIKE 1 (ERL1), and ERECTA-LIKE 2 (ERL2)[21]. Mutations in ER genes result in a dwarf phenotype, blunt siliques, and compact inflorescence architectures, first identified in the *Arabidopsis Landsberg erecta* (Ler) ecotype[22,23]. The ERf genes also act redundantly in regulating SAM development. Single erf mutants do not have significant alterations in SAM size, whereas er erl1 erl2 triple mutants have enlarged meristems[24–26]. ER ligands, including EPFL1, EPFL2, EPFL4, and EPFL6, act non-cell autonomously, and also act redundantly to regulate SAM size and inflorescence architecture. High-order mutants in these ligands have wider SAMs, reduced stature, and compact inflorescences[18–20]. Genetic analyses suggest an interconnection between arabidopsis ER and CLV-WUS pathways, since ERfs and CLV3 synergistically regulate SAM size. Moreover, wus mutations are epistatic to ERf genes, and EPL signaling suppresses CLV3 and WUS expression[27–30]. Several studies also highlight the involvement of plant hormones, including cytokinin and auxin, in ERf-EPFL-mediated meristem and inflorescence development[20,25,26]. The role of the ER pathway in meristem development in cereal crops is not fully elucidated, though in rice, it has been implicated in regulating panicle architecture by modulating cytokinin metabolism through a mitogen-activated protein kinase cascade[31]. Notably, optimization of rice panicle architecture has been achieved by specifically suppressing OsEPFL-OsER1 ligand–receptor pairs[32]. Similarly, targeted genetic modifications of ER and EPFL genes have been used to customize plant architecture in tomato, further underscoring the potential of this pathway for crop improvement[33,34]. Despite these advances, the functions of ERECTA pathways in the key cereal crop maize, particularly in regulating meristem development and yield traits, have not been explored.

Here, we characterized the role of the three maize ERECTA (ZmER) family genes, ZmER1, ZmER2, and ZmERL, in regulating meristem size as well as plant and ear architecture. Mutations in Zmer1 but not Zmer2 or Zmerl resulted in compact plant architecture, enlarged SAMs and IMs, and fasciated ears. High-order mutants involving Zmer2 and Zmerl further enhance the Zmer1 phenotype, revealing functional redundancy among family members and a primary role of ZmER1. Using alphafold predictions and microscale thermophoresis (MST) assays, we found that ZmER1 binds specifically to five EPFL peptides. Double and triple mutant combinations containing Zmepfl1-1 exhibited enlarged IMs, and the quintuple mutants display the most pronounced

IM fasciation, suggesting that these peptides may act redundantly in IM development. Additionally, we found that ZmWUS1 is upregulated in Zmer mutants, and the enlarged IM in Zmer1 mutants is significantly suppressed by mutations in Zmwus1. Finally, we identified weak Zmer1 alleles that improved agronomic traits such as leaf angle and KRN, highlighting the potential to modulate ZmER activity for yield trait improvement and high-density planting.

## Results

### CRISPR/Cas9 knockouts of *ZmERs* resulted in compact plant architecture and altered shoot apical meristem size

Two co-orthologs of arabidopsis ER, ZmER1 and ZmER2 were present in the maize genome, along with a single ZmERL ortholog related to arabidopsis ERL1 and ERL2 (Supplementary Fig. 1)[35]. To explore the function of these genes, we generated knockout alleles using the CRISPR-CAS9 system with multiplexed guides (Fig. 1a). Transgenic plants carrying various mutant alleles were backcrossed three times with the B73 inbred line to segregate out the Cas9 transgene and to standardize the genetic background. Edited plants were then crossed and selfed to produce single, double and triple mutants. We found that Zmer1 mutants had a compact architecture, characterized by reduced plant and ear height, shortened internodes and decreased leaf angle, leaf width, and leaf length (Fig. 1b, c, Supplementary Figs. 2–5). In contrast, the Zmer2 and Zmerl single mutants, as well as the Zmer2;Zmerl double mutants, had no obvious changes in plant height or leaf size (Fig. 1b, c, Supplementary Figs. 2, 4). Interestingly, the mutation of either Zmer2 or Zmerl significantly enhanced the architecture phenotype of the Zmer1 mutants. The Zmer1;Zmer2 and Zmer1;Zmerl double mutants were more compact, with further reduction in plant and ear height, as well as narrower and shorter leaves, compared to the Zmer1 single mutants (Fig. 1b-c and Supplementary Fig. 4). Internode length and cell size were further reduced in Zmer1;Zmer2 compared with Zmer1, consistent with the greater reduction in plant height (Supplementary Fig. 5). Additionally, the Zmer triple mutant has the most severe phenotype (Fig. 1b, Supplementary Fig. 4a, c). Plant and ear height, as well as the leaf width, were statistically reduced compared to the Zmer1;Zmer2 and Zmer1;Zmerl double mutants, as determined by Student t-test analysis (Fig. 1c, Supplementary Fig. 4b, e). These results indicate that ZmER genes redundantly regulate plant architecture, with ZmER1 playing the primary role.

To investigate whether the altered plant architecture in Zmer mutants were associated with shoot apical meristem development, we measured SAM width and height in single and double mutants. The Zmer1 single mutant had increased SAM width and reduced SAM height compared to WT, while the Zmer2 single mutant showed no significant changes in SAM size (Fig. 1d–f). This observation aligns with the finding that Zmer1 but not Zmer2 have vegetative architecture phenotype (Fig. 1b, c, Supplementary Fig. 4). Additionally, the Zmer1;Zmer2 double mutants have an even greater increase in SAM width compared to the Zmer1 single mutant (Fig. 1d–f). In summary, our results indicate that ZmER1 and ZmER2 genes function redundantly, though unequally, in regulating SAM development, with ZmER1 playing a primary role.

To further investigate the function of ZmERs, we checked their expression patterns in SAM from the laser micro-dissected cell RNA-seq profiling data[8]. All three ZmERs were expressed in distinct functional domains of the shoot apex, including meristem, epidermal layer of dividing cells (L1), the incipient primoridium within the SAM (P0), the first visible leaf is a plastochron 1 (P1), plastochron 2 (P2), plastochron 3 (P3) and internodes. ZmER1 had the highest expression, and ZmER2 and ZmERL were significantly lower[8] (Supplementary Fig. 6). This observation aligns with our finding that ZmER1 plays a more prominent role compared to the other two genes. To validate the expression pattern of ZmER1, we generated a translational fusion of the ZmER1 genomic sequence with a miniTurbo-YFP at its C terminus,

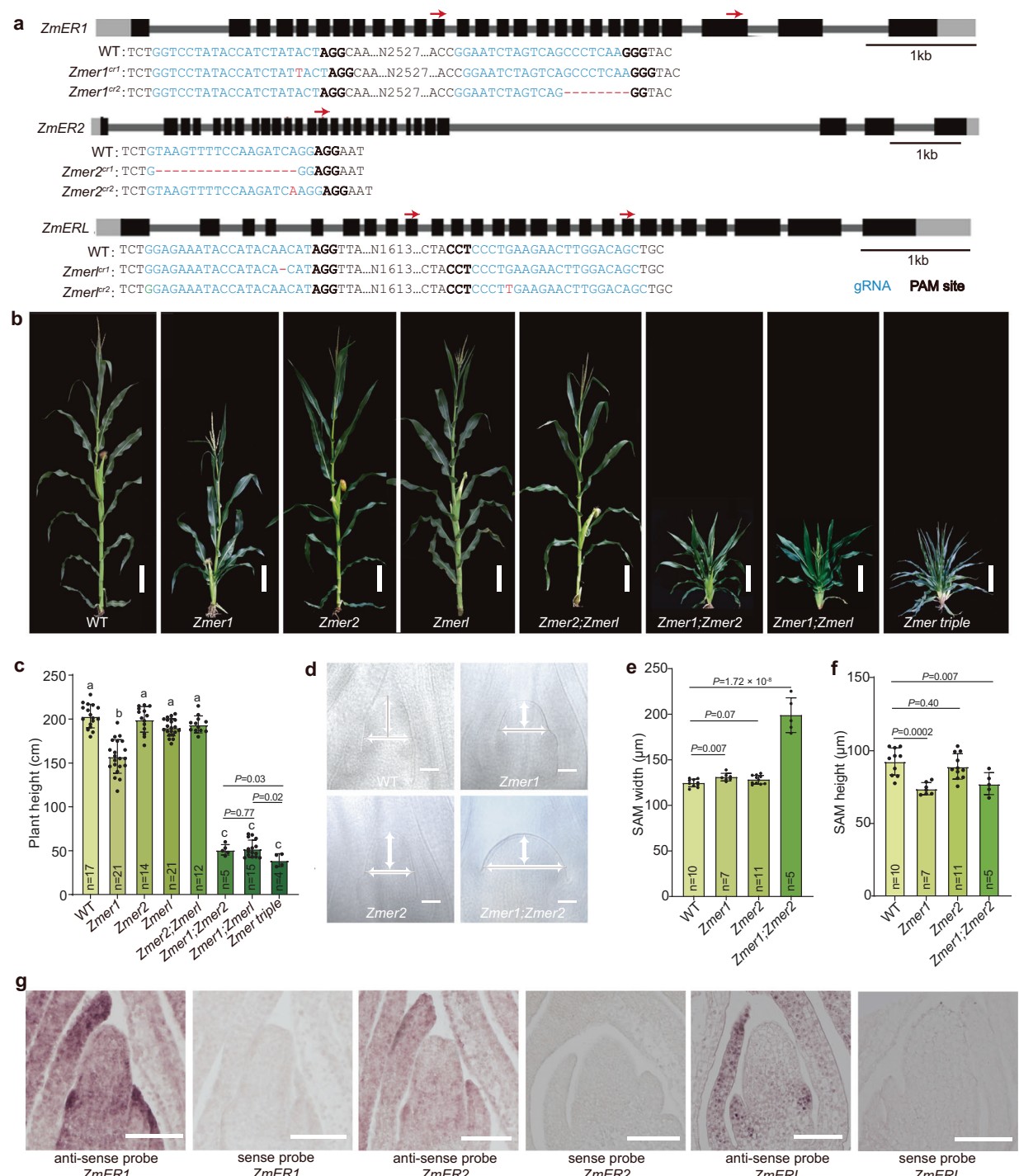

**Fig. 1 | The *Zmer* mutants have compact plant architecture and larger shoot apical meristems. a** CRISPR/Cas9 editing generated different frameshift alleles for the *Zmers*. The positions of guide RNAs are indicated by red arrows on the gene models. **b** Representative images of mature maize plants illustrate that mutations in *ZmER1* and its paralogs significantly reduce plant height. **c** Statistic analysis of plant height in various *Zmer* mutants. Different letters indicate statistically significant groups at *P* < 0.05 (one-way ANOVA with post hoc Tukey's multiple comparison test). Although the *Zmer1;Zmer2*, *Zmer1;Zmerl* and the triple mutants are classified within the same group, the triple mutant is significantly shorter than double mutants as determined by a two-sided unpaired Student *t*-test. The exact *p*-values are shown in the figure. **d** Images of representative cleared SAMs, meristem height and width are marked by the white arrows. *Zmer1* single mutants, and *Zmer1;Zmer2* double mutants have increased SAM width and reduced SAM height as quantified in (**e** and **f**). **g** In situ hybridization analysis shows *ZmER1*, *ZmER2*, and *ZmERL* are expressed in the SAM, with stronger signals in the leaf primordia. Each experiment is independently repeated three times, yielding consistent results. n represents the number of samples (**c**, **e**, and **f**). Scale bars, 20 cm (**b**), 50 μm (**d**) and 100 μm (**g**). For **e** and **f**, data are presented as means ± s.d. *p*-values are calculated using a two-sided *t*-test, for **b**–**f**, phenotypic analyses use the *cr1* alleles for all three genes. Superscript "cr1" is omitted for space. Source data for statistical analysis in **c**, **e**, and **f** are available in the Source Data file.

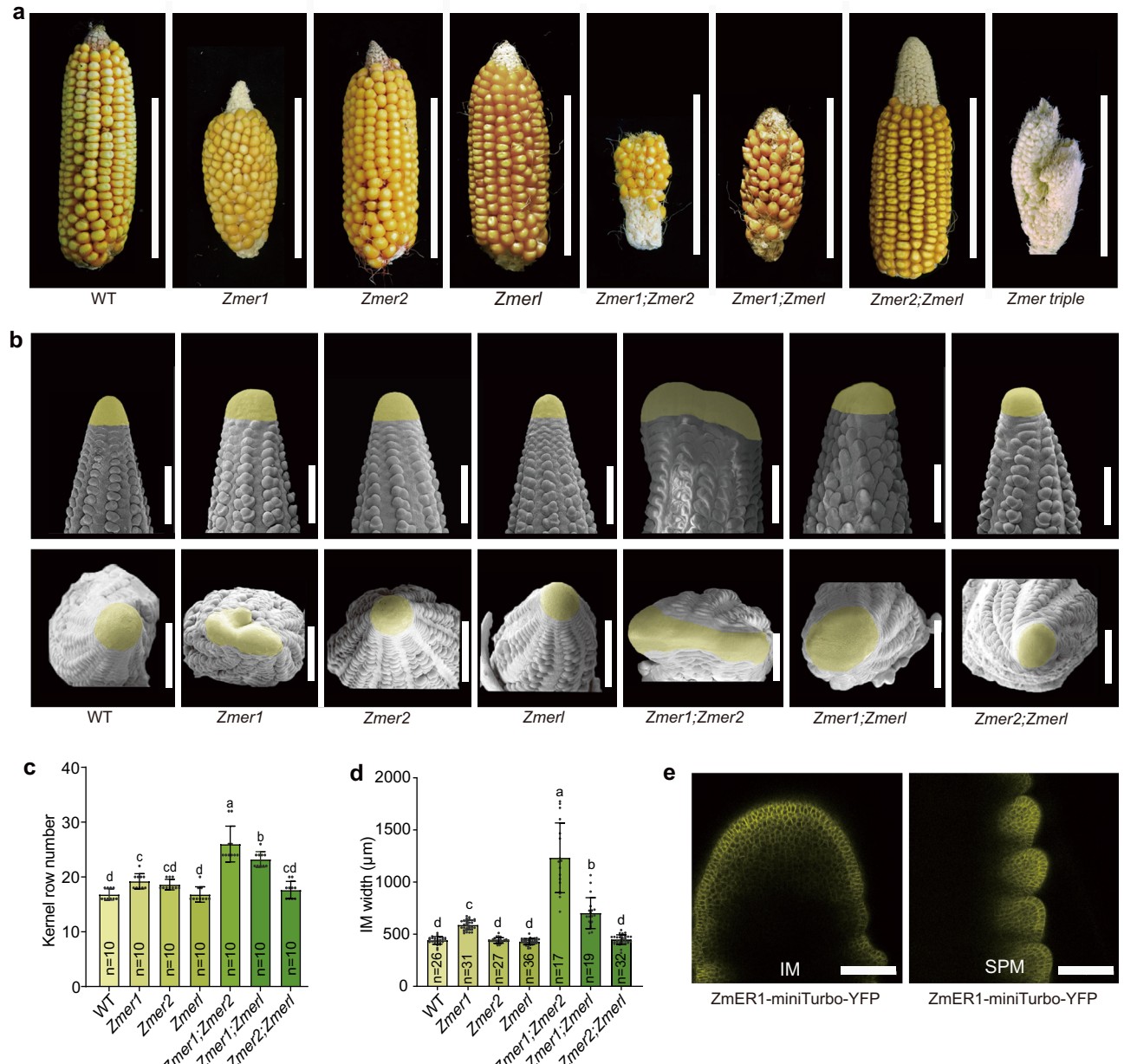

**Fig. 2 | *ZmER* genes are essential for ear and inflorescence meristem development. a** *Zmer1* single mutants, *Zmer1;Zmer2*, and *Zmer1;Zmerl* double mutants and *Zmer* triple mutants have fasciated ears with increased kernel row number compared to wild-type (WT), as quantified in (**c**). **b** Side and top-down scanning electron microscopy views of ear primordia. The width of the IM is increased in *Zmer1* single, *Zmer1;Zmer2*, and *Zmer1;Zmerl* double mutants, with a particularly significant increase observed in the *Zmer1;Zmer2* double mutants as quantified in (**d**). **e** ZmER1-miniTurbo-YFP expression is detected throughout the inflorescence meristem (IM) and spikelet pair meristem (SPM). Each experiment was independently repeated three times with similar results. Scale bars, 10 cm (**a**), 500 μm (**b**) and 100 μm (**e**). Data in **c**, **d**, are presented as means ± s.d. (one-way ANOVA with post hoc Tukey's multiple comparison test, *P* < 0.05). Source data for the statistical analyses in **c**, **d** are provided in the Source Data file.

under the control of its native promoter and terminator (Supplementary Fig. 7a). This construct was transformed into maize and backcrossed twice with the *Zmer1* mutant. ZmER1-miniTurbo-YFP successfully complemented the dwarf phenotype of *Zmer1* mutants, indicating that the transgene was functional (Supplementary Fig. 7b, c). Confocal imaging revealed ZmER1-miniTurbo-YFP expression throughout the SAM, and the protein was localized primarily on the membrane (Supplementary Fig. 7d). In addition, in situ hybridization experiments also showed that *ZmER1*, *ZmER2* and *ZmERL* were all expressed in the SAM, with stronger signals in leaf primordia (Fig. 1g).

## ZmER genes are required for ear and inflorescence meristem development

In addition to the vegetative architecture defects, the *Zmer1* mutants also had fasciated ear phenotypes and an increased kernel row number (KRN) (Fig. 2a, c). In contrast, the *Zmer2* and *Zmerl* single mutants, as well as the *Zmer2;Zmerl* double mutant, maintained normal ear architecture (Fig. 2a). However, *Zmer2* significantly enhanced the ear defects caused by *Zmer1*, as the *Zmer1;Zmer2* double mutants had more strongly fasciated ears and a dramatic increase in KRN. *Zmerl* mutants also enhanced *Zmer1*, but to a lesser extent (Fig. 2a, c). *Zmer* triple mutants had the most pronounced fasiated ears compared to

the single and double mutants (Fig. 2a). These results suggest that *Zmer2* and *Zmerl* function redundantly but unequally with *Zmer1* in regulating ear architecture.

We further investigated the roles of *ZmER* genes in inflorescence meristem (IM) development. Consistent with the ear phenotypes, *Zmer1* single mutants have larger IMs compared to WT, while the *Zmer2* and *Zmerl* single mutants, as well as the *Zmer2;Zmerl* double mutant, showed no change. Both the *Zmer1;Zmer2* and *Zmer1;Zmerl* double mutants had significantly larger IMs compared to the *Zmer1* single mutants, with the IMs of *Zmer1;Zmer2* being larger than *Zmer1;Zmerl* (Fig. 2b, d). In summary, our results indicate that *Zmer* genes function redundantly but unequally in regulating ear architecture and IM development, with *Zmer1* playing a primary role and *Zmer2* having a stronger effect than *Zmerl*. This differs from their role in regulating plant architecture and vegetative SAM development, where *Zmer1* plays the primary role, and *Zmer2* and *Zmerl* contribute similarly.

RNA-seq profiling data revealed that while all three *ZmER* were expressed in the inflorescence meristem (IM), spikelet pair meristem (SPM), spikelet meristem (SM), and floral meristem (FM), ZmER1 had significantly higher expression compared to the other two genes, consistent with its primary role in the IM regulation[36] (Supplementary Fig. 8). Confocal imaging of ear primordia from the ZmER1-miniTurbo-YFP plants further validated that ZmER1 was highly expressed in IM and SPM (Fig. 2e).

## Five EPFL peptides function as ZmER1 ligands

Studies in other plant species, such as arabidopsis and rice found that the ERECTA protein can bind different EPIDERMAL PATTERNING FACTOR (EPF) or EPF-like peptides to regulate various biological processes[18,20,32,37–40]. Phylogenetic analysis revealed that the maize genome encodes 3 EPF and 13 EPFL peptides (Supplementary Fig. 9)[41]. To ask whether EPFL/EPF peptides regulate shoot and inflorescence architecture, we selected six *EPFL* genes that were relatively highly expressed in the SAM and IM, based on the available RNA seq data[8,42] (Supplementary Fig. 9). We expressed Cas9 with six single-guide RNAs (sgRNAs) targeting the six *EPFL* genes. After transformation into the maize inbred line KN5585, successful editing was identified for five genes by PCR amplification and Sanger sequencing; however, no edits were detected in ZmEPFL4-1 (Fig. 3a). Lines carrying mutations in different *ZmEPFL* genes were backcrossed twice with the KN5585 inbred line to segregate away off-target mutations and remove the Cas9 transgene to avoid further edits. With the mutant combinations obtained, the single mutants *Zmepfl1−2*, *Zmepfl2-1*, and *Zmepfl4-2*; the double mutant *Zmepfl1-2;Zmepfl4-2* and *Zmepfl2-1;Zmepfl4-2* and the triple mutant *Zmepfl1-2;Zmepfl2-1;Zmepfl4-2* and *Zmepfl1-2;Zmepfl4-2;Zmepfl9-1*, have normal IM development (Fig. 3c). By contrast, the double mutants *Zmepfl1-1;Zmepfl1-2* and *Zmepfl1-1;Zmepfl9-1*, as well as the triple mutants *Zmepfl1-1;Zmepfl1-2;Zmepfl4-2* and *Zmepfl1-1;Zmepfl1-2;Zmepfl9-1*, exhibit noticeable IM fasciation (Fig. 3c). These results suggest that *ZmEPFL1-1* plays a relatively predominant role in regulating meristem size, as combinations including *Zmepfl1-1* displayed fasciated IMs, whereas other double and triple combinations did not. Notably, the quintuple mutant displayed the most pronounced IM fasciation among all genotypes examined, which may reflect partial functional redundancy among these ZmEPFL peptides (Fig. 3). No mutant combinations including the quintuple mutant showed obvious changes in overall plant architecture (Fig. 3), implying that the *ZmEPFL* genes may primarily influence inflorescence meristem development. Nonetheless, further work involving all possible mutant combinations will be required to fully clarify the specific functions and genetic relationships of individual *ZmEPFL* genes in plant and meristem development. Consistent with the enlarged IMs, the *Zmepfl* quintuple mutants also had strongly fasciated ears and a significantly higher kernel row number (Fig. 3d, e).

To further investigate whether ZmER functions as a receptor for the selected EPFL peptides, we employed microscale thermophoresis (MST), a technique that quantifies biomolecular interactions based on thermophoresis. The MST assay found that ZmER1 can indeed bind to the five EPFL peptides, albeit with varying binding affinities (Fig. 4a–f). Specifically, ZmEPFL4-2 had the strongest binding affinity, as indicated by the lowest Kd value, while ZmEPFL9-1 had the weakest binding affinity (Fig. 4a–f). As a control, the CLE peptide ZmFCP1 was not able to bind to ZmER1, further confirming the specificity of ZmER1 for its ligands (Fig. 4f). Consistently, the interaction between ZmER1 and four EPFLs were also predicted by AlphaFold multimer, with high ipTM values exceeding 0.71 (Fig. 4g–j). Among these, the interaction between ZmER1 and ZmEPFL4-2 was predicted with the highest ipTM value of 0.88, aligning with the strongest binding affinity observed in the MST experiment.

## The function of ZmER1 in inflorescence meristem development relies on ZmWUS1

To further understand how ZmERs affect the meristem development, we carried out RNA seq analysis of WT, *Zmer1;Zmer2* and *Zmer1;Zmerl* using IMs dissected from 2-5 mm ear primordia. The RNA profiling identified 1,265 up-regulated differentially expressed genes (DEGs) and 303 down-regulated DEGs in *Zmer1;Zmer2* and 1421 up-regulated DEGs and 347 down-regulated DEGs in *Zmer1;Zmerl* (Supplementary Data 2 and 3, Fig. 5a) (DEGs; P-*value* (padj) <0.05 and log₂(fold change) >1). Approximately 1/3 of the DEGs were similarly differentially expressed in both double mutants, revealing a common transcriptional change in *Zmer* mutants (Fig. 5a). Gene Ontology (GO) enrichment analysis revealed that DEGs in both mutants were enriched for biological processes related to hormone regulation and response, as well as developmental regulation (Supplementary Fig. 10). Consistent with this, a set of genes involved in regulation of development, meristem activity and hormone biosynthesis and signaling were significantly differentially expressed in both mutants (Fig. 5b, c). Notably, *ZmWUS1*, the core transcription factor of the CLAVATA pathway[43,44], was commonly up-regulated in both *Zmer1;Zmer2* and *Zmer1;Zmerl* (Fig. 5b, c). Besides, two *CLE* genes, *ZmFCP1* and *CLE21*, which are putatively *ZmWUS1* responsive, were also upregulated. The up-regulation of *ZmWUS1* in the inflorescence meristem in the *Zmer* mutant was further verified by RT-qPCR (Fig. 5d, Supplementary Fig. 11). This result aligns with previous studies that ER constrains the expression of *WUS* in arabidopsis[29].

To further investigate whether ZmWUS1 functions downstream of ZmERs, we carried out genetic analysis between *Zmer1* and *Zmwus1*. An early stop codon-gained mutant of *Zmwus1* was isolated from an EMS mutant stock (Supplementary Fig. 12)[45]. The *Zmwus1* mutants didn't have any obvious altered plant architecture or inflorescence meristem defects (Fig. 5e–i), consistent with CRISPR induced knock-out alleles described in a previous study[43]. Nevertheless, *Zmwus1* mutation significantly suppresses the IM fasciation caused by the *ramosa1 enhancer locus 2* (*rel2*) and *rel2;rel2-like1* (*relk1*) mutants, revealing a key function of ZmWUS1 in meristem development[46]. Interestingly, the compact plant architecture of *Zmer1* was not suppressed by *Zmwus1*, as the *Zmer1;Zmwus1* double mutant showed similar plant and ear height to the *Zmer1* single mutant (Fig. 5e–g). In contrast, the *Zmwus1* mutation significantly suppressed the enlarged IM phenotype of *Zmer1*, with the IM size of *Zmer1; Zmwus1* double mutant being significantly smaller than that of *Zmer1* (Fig. 5h–i). These findings suggest that *ZmWUS1* is required for the enlarged meristem phenotype of *Zmer1*, whereas the compact plant architecture of *Zmer1* appears largely *ZmWUS1*-independent. Moreover, these results also suggest that IM development can be uncoupled from the plant architecture, consistent with previous studies showing that variations in *Kernel Row Number 2* (*KRN2*) and *KRN4* lead to enlarged IMs while maintaining normal plant architecture[47,48].

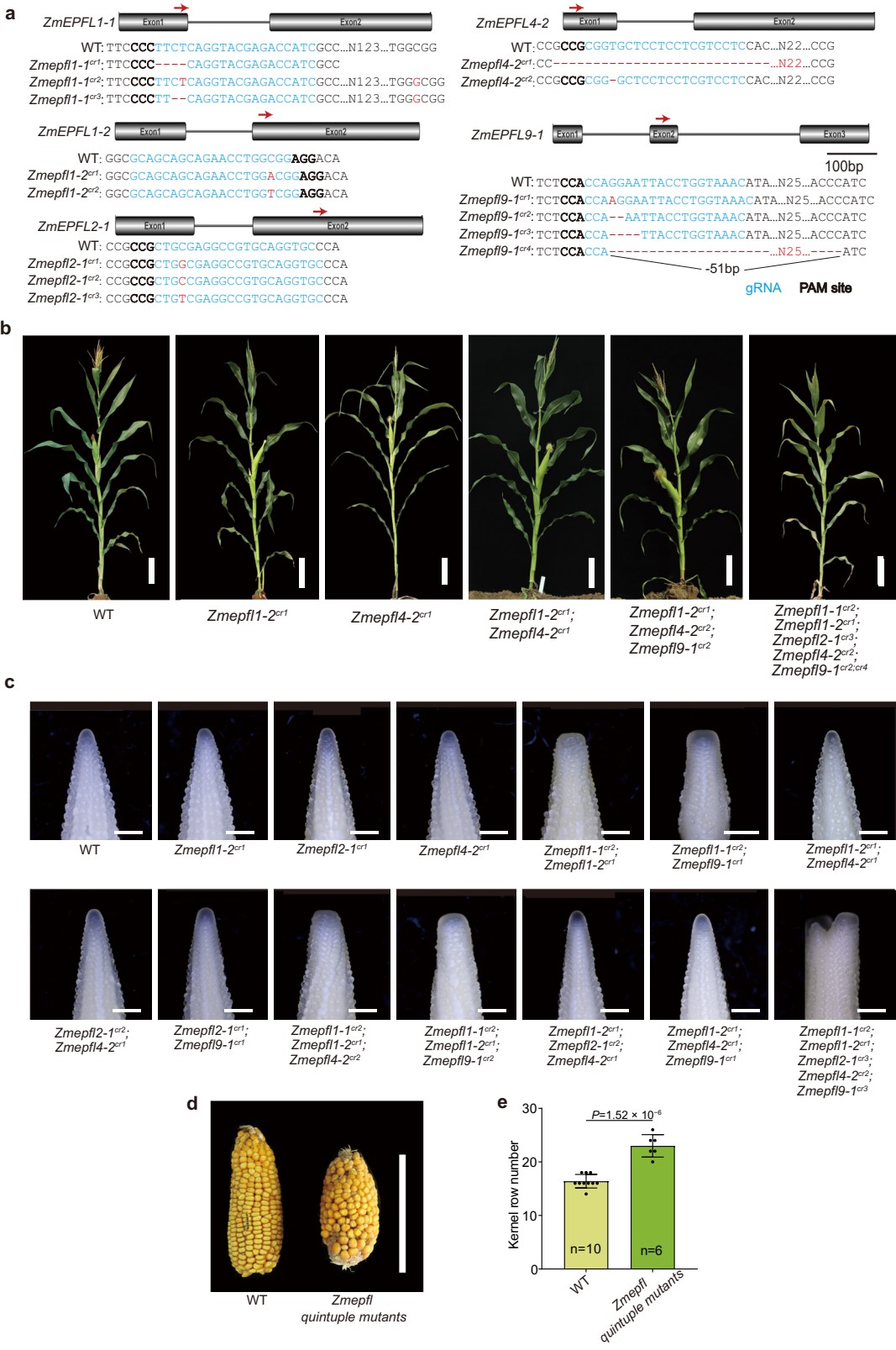

**Fig. 3 | *Zmepfl* mutants redundantly regulate the development of inflorescence meristems. a** CRISPR/Cas9 editing generated different, distinct frameshift alleles in five *EPFL* genes. **b** Representative images of overall plant architecture in the indicated mutant combinations. **c** Images of young ear primordia showing the IM development in different mutant combinations. **d** The quintuple *Zmepfl* mutants

have fasciated ears. **e** Statistical analysis shows that the kernel row number is significantly increased in the quintuple mutants. n represents the number of samples. Scale bars, 20 cm (**b**), 500 μm (**c**), and 10 cm (**d**). For **e**, data are presented as mean values ± s.d., from a two-tailed, two-sample *t*-test. The source data underlying the statistical analysis in **e** are provided in the Source Data file.

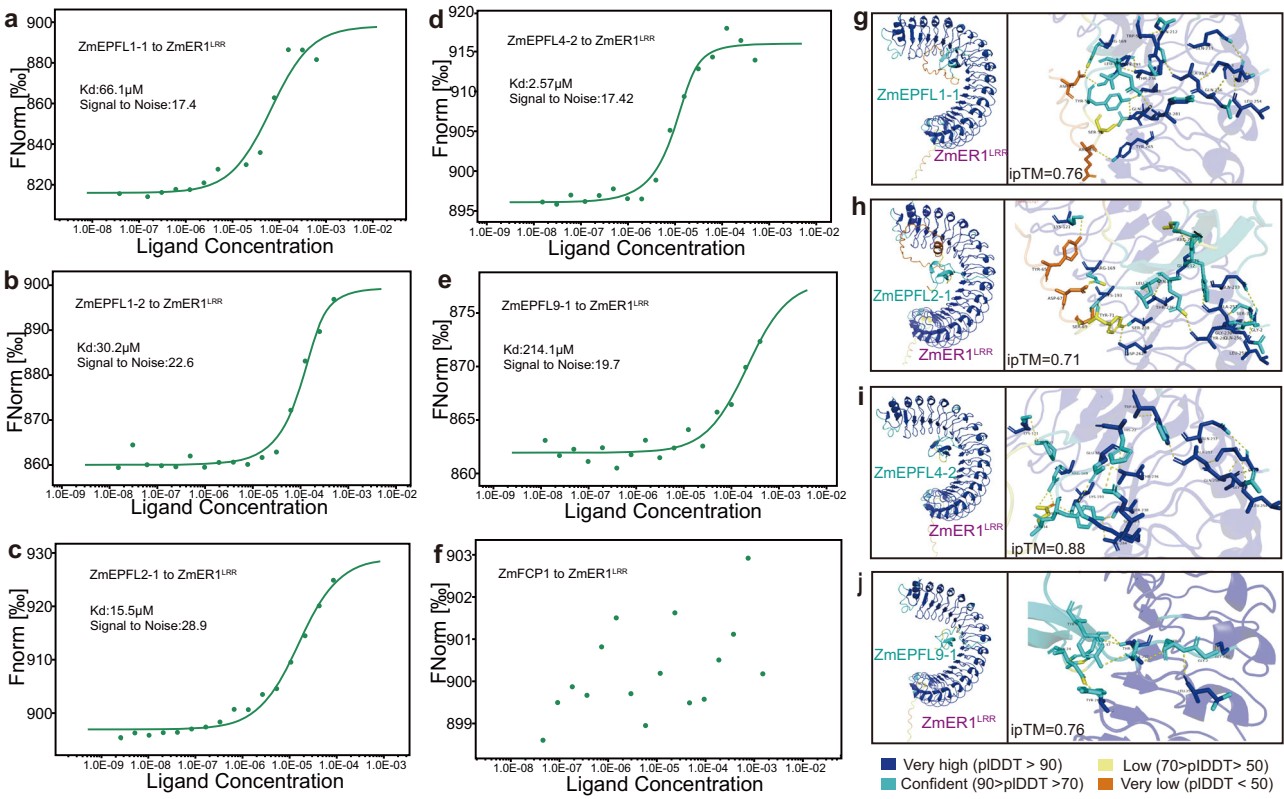

**Fig. 4 | ZmEPFL ligands directly bind the ZmER1 receptor. a–f** Quantification of the binding affinity of ZmER1$^{LRR}$ and ZmEPFL1-1 (**a**), ZmEPFL1-2 (**b**), ZmEPFL2-1 (**c**), ZmEPFL4-2 (**d**), ZmEPFL9-1 (**e**) and ZmFCP1 (**f**) as measured with microscale thermophoresis (MST) method. GST-tagged ZmER1$^{LRR}$ is fluorescently labeled. The signal-to-noise and dissociation constant (Kd) are indicated. Each experiment was repeated two to three times with similar results. Raw data are provided in the Source Data file. **g–j** Molecular docking between ZmER1$^{LRR}$ and ZmEPFL1-1 (**g**), ZmEPFL2-1 (**h**), ZmEPFL4-2 (**i**), ZmEPFL9-1 (**j**). Protein interaction predictions are performed using the Alphafold server, with input data for the ZmER1$^{LRR}$ protein and the mature ZmEPFL peptide. In the predicted interaction network, ipTM measures the accuracy of the predicted relative positions of subunits within the complex. pLDDT: a per-atom confidence estimate on a 0-100 scale, where higher values indicate greater confidence. The pLDDT is displayed as color outputs in the image of the structure. The predicted interaction sites are visualized and annotated using the PyMOL server.

## Weak alleles of *ZmER1* improve yield traits

The null mutants of *Zmer1* have a compact plant architecture with reduced plant height and decreased leaf angle while their ears are fasciated with an increased kernel row number (Fig. 1b–c, Fig. 2a, c and Supplementary Fig. 3). While these traits highlight the potential of the ZmER1 locus for both high-density planting and yield improvement, the phenotype of the null alleles might be too strong to be practical. To mine additional alleles with potential benefits for plant architecture and yield, we identified two nonsynonymous alleles from an EMS mutant library[45] (Supplementary Fig. 13). These alleles were designated as *Zmer1$^{D295N}$* and *Zmer1$^{A369T}$* based on their respective amino acid changes. Unlike the *Zmer1* loss-of-function mutant, which had a significant reduction in plant height, neither of these two alleles had a notable plant height decrease (Fig. 6a, b). Interestingly, however, the *Zmer1$^{D295N}$* and *Zmer1$^{A369T}$* had significantly reduced leaf angles, like the *Zmer1 null allele* mutant (Fig. 6c, e). Reduced leaf angle is considered a key trait for optimizing maize architecture for dense planting. Therefore, these two alleles showed favorable agricultural traits for high-density planting. In addition, *Zmer1$^{D295N}$* and *Zmer1$^{A369T}$* also had mild ear fasciation, without the noticeable reduction in ear length observed in the *Zmer1$^{cr1}$* mutant (Fig. 6f). Consistent with the mild fasciation phenotype, the *Zmer1$^{A369T}$* allele significantly increased kernel row number, and *Zmer1$^{D295N}$* showed a trend toward an increase, though not statistically significant (Fig. 6g). And, both alleles didn't have negative effect on the kernel number per row, ear length, or hundred-kernel weight, which are often reduced in strong fasciation mutants (Fig. 6h, j). These findings suggest that *Zmer1$^{D295N}$* and *Zmer1$^{A369T}$* function as weak alleles, with both alleles significantly improving leaf angle, and *Zmer1$^{A369T}$* also enhancing ear architecture.

## Discussion

Maize is a globally important crop, and its yield is a complex trait influenced by the combined contributions of yield per plant and plant density per unit area. Yield per plant largely depends on ear architecture, which is shaped by kernel row number (KRN) and kernel number per row (KNR)[4,14,47,49]. Both traits are heavily reliant on inflorescence meristem activity to produce sufficient spikelet pairs and spikelets that form kernels. On the other hand, planting density is strongly influenced by the above-ground plant architecture, which affects light harvesting, lodging resistance, and plant competition[2,50–52]. Manipulating plant height and leaf angle are effective strategies to improve plant density to increase yield[53–55]. Here, we showed that the three ERECTA family members, ZmER1, ZmER2, and ZmERL play pleiotropic roles in regulating several key agronomic traits. *Zmer* mutants had increased KRN, reduced plant, ear height and decreased leaf angles, which are all beneficial phenotypes toward enhancing yield and increasing plant density (Figs. 1, 2, Supplementary Figs. 3, 4 and Fig. 6). However, the null alleles of *Zmers* might not be practical for yield improvement due to largely reduced plant and ear height and strong fasciation with disordered kernel rows. In this study, we showed that two weak alleles of *Zmer1* exhibited variable beneficial effects on plant and ear architecture, highlighting the potential to

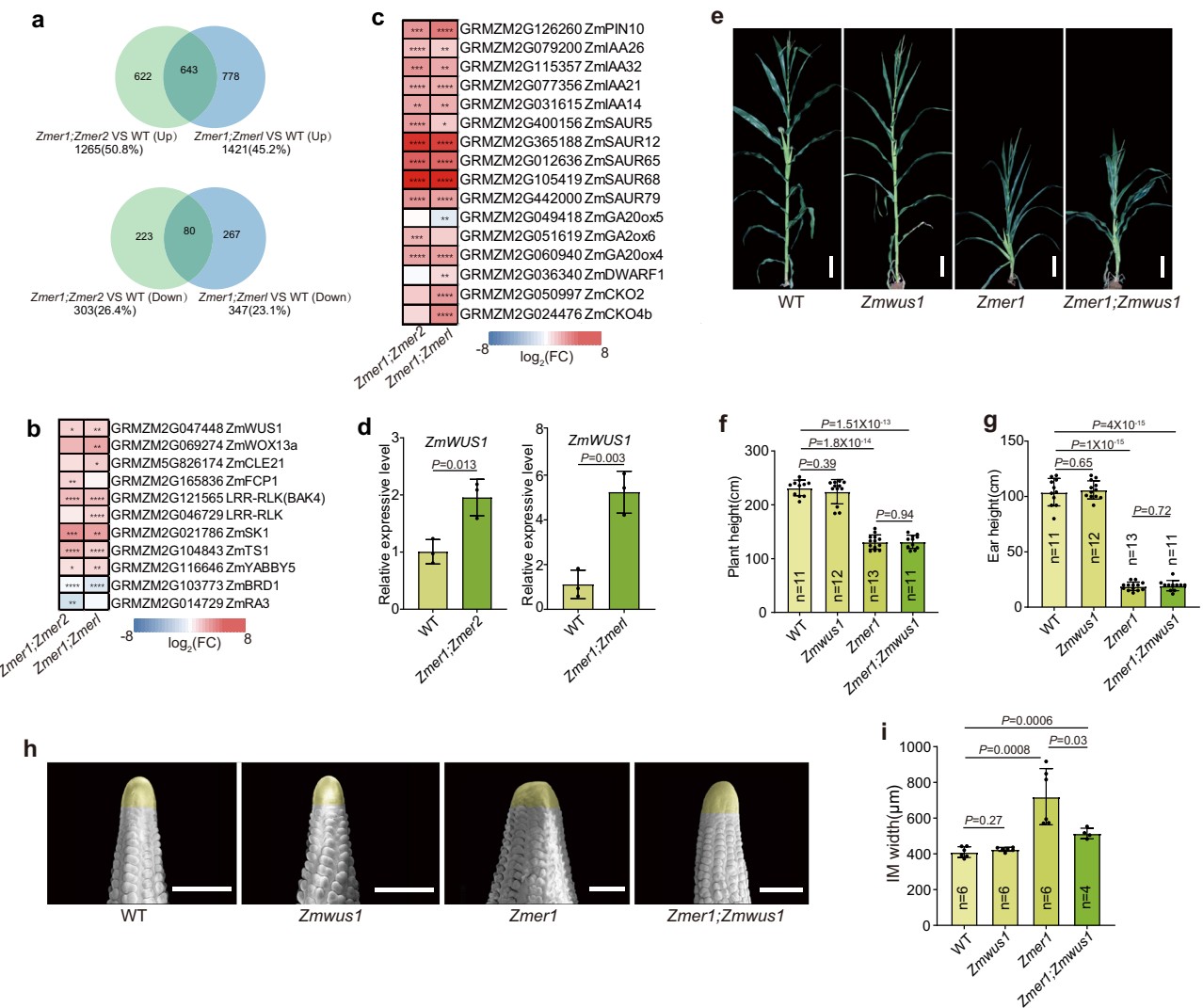

**Fig. 5 | ZmER1 function in inflorescence meristem development relies on ZmWUS1. a** Differentially expressed genes (DEGs) in *Zmer1;Zmer2* and *Zmer1;Zmerl* double mutants show significant overlap. The top panel displays the number of upregulated DEGs identified in *Zmer1;Zmer2* compared to WT, in *Zmer1;Zmerl* compared to WT, and their overlap. The bottom panel similarly highlights downregulated DEGs. **b** Heatmap illustrating DEGs in *Zmer1;Zmer2* and *Zmer1;Zmerl* mutants associated with the key developmental regulators and hormone signaling and biosynthesis (**c**). Differential expression is analyzed using DESeq2 with the Wald test (two-sided), and *p*-values are adjusted for multiple comparisons using the Benjamini–Hochberg false discovery rate (FDR) method. Significance is indicated as: * padj <0.05, ** padj <0.01, *** padj <0.001, **** padj <0.0001 with log2 (FC) > 1. **d** RT-qPCR analysis verifies the significant upregulation of *ZmWUS1* in the IM of *Zmer1;Zmer2* and *Zmer1;Zmerl* mutants compared to WT. Data are presented as

mean values ± s.d. with statistical significance assessed using a two-tailed Student's *t*-test. Three independent biological replicates are performed, each including three technical replicates. Approximately 10 IMs per genotype are used for each replicate. **e** *Zmer1;Zmwus1* double mutants have similar height as *Zmer1* single mutants. **f, g** Statistical analyses revealed that there were no significant differences in plant height or ear height between the *Zmer1;Zmwus1* double mutant and the *Zmer1* single mutant. **h** SEM images of ear primordia from WT, *Zmer1*, *Zmwus1* and *Zmer1;Zmwus1* reveal that *Zmwus1* mutants suppress the enlarged IM of *Zmer1*. **i** Quantification of IM size of WT, *Zmer1*, *Zmwus1* and *Zmer1;Zmwus1* shows that *Zmer1* significantly suppresses the enlarged IM of *Zmer1*. Data are presented as the means ± s.d. The sample size (n) and *p*-values for each group are labeled in the figure. *p*-values are calculated using two-tailed Student's *t*-tests (**f**, **g**, and **i**). Scale bars, 20 cm (**e**), 500 μm (**f**).

improve yield under high-density planting condition by genetically manipulating the *ZmER1* locus. Nevertheless, their effect should be further evaluated through large-scale yield tests with commercial cultivars, diverse plant conditions, and multiple environments to better assess the impact of these genetic variations on yield traits[56].

Compared to the eudicot arabidopsis, which contains one ERECTA gene and two ERL genes, the monocot maize contains two ERECTA genes and one ERL gene[35], suggesting a diversification of ERECTA families in different species (Supplementary Fig. 1). Our genetic analyses indicate that ZmER1 plays the primary role in regulating meristem development and plant architecture, with ZmER2 and ZmERL providing partially redundant functions. This genetic

relationship aligns with that observed in arabidopsis, where *er* mutants display compact shoots and inflorescences, and *erl1* and *erl2* act redundantly to enhance the *er* phenotypes[25,26]. However, in maize, ZmER2 appears to contribute more strongly than ZmERL to inflorescence meristem (IM) size and ear architecture, as *Zmer1; Zmer2* double mutants exhibit more severe fasciation than *Zmer1;Zmerl* plants, consistent with the closer phylogenetic relationship between ZmER1 and ZmER2 (Fig. 2). This differs slightly from arabidopsis, where *erl1* and *erl2* enhance the *er* phenotype in partially overlapping ways, but no clear functional hierarchy between them has been established[21]. Notably, ZmER1 appears to have a more pronounced effect on meristem size than its arabidopsis counterpart. *Zmer1* single mutants

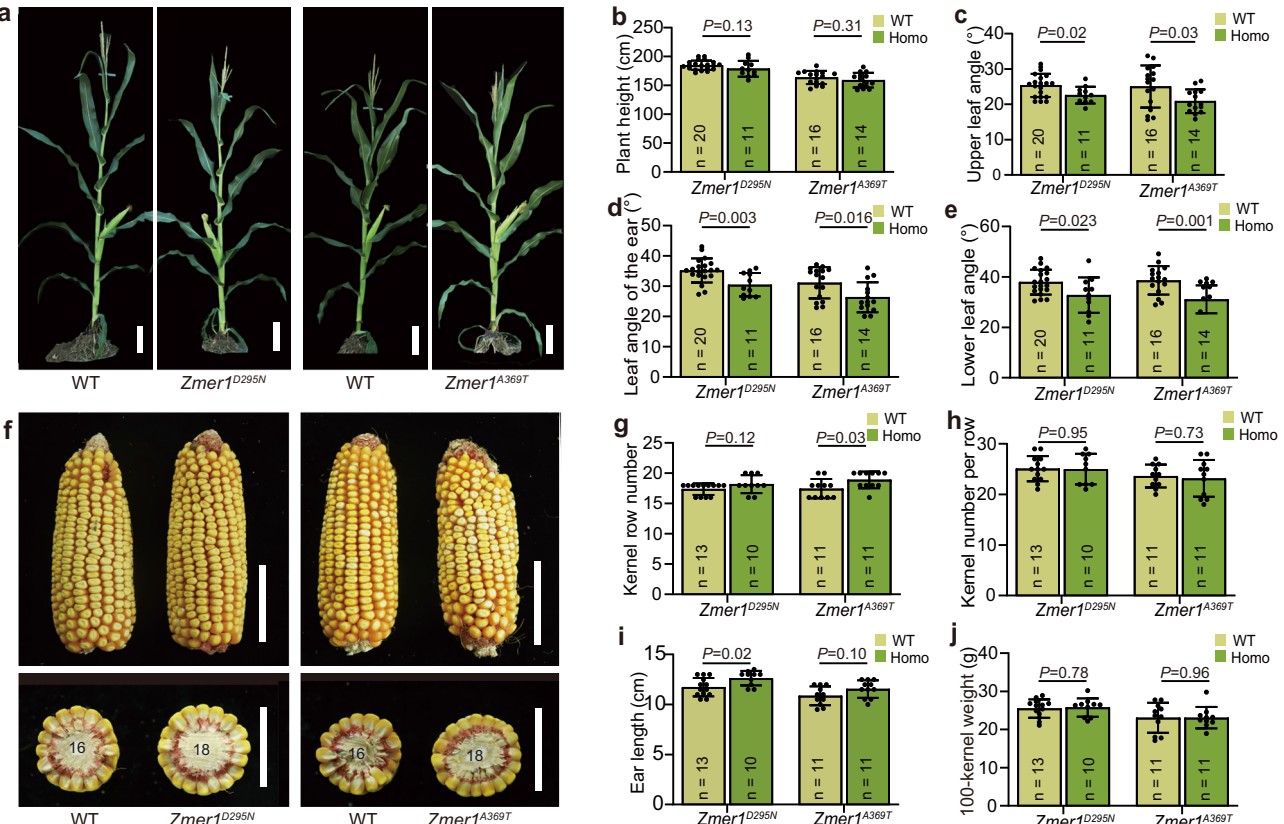

**Fig. 6 | Weak alleles of *Zmer1* reduce leaf angle and increase kernel row number.** **a** Representative images of the two *Zmer1* weak alleles, and their respective WT siblings. **b–e** Statistical analysis of plant height and leaf angle for the weak *Zmer1* alleles compared to their WT siblings. *Zmer1^{D295N}* and *Zmer1^{A369T}* mutants have reduced leaf angles. **f** Representative mature ears from the two *Zmer1* weak alleles, and their corresponding WT siblings. **g–j** Statistical analysis of kernel row number (**g**), kernel number per row (**h**), ear length (**i**) and hundred-kernel weight (**j**). The sample size (n) and *p*-values for each group are labeled in the figure, *p*-values are calculated using two-tailed Student's *t*-tests (**b–e**, **g–j**). Scale bars, 20 cm (**a**) and 5 cm (**f**).

already display significantly enlarged SAMs and IMs, and meristem size is further increased in *Zmer1* double and triple mutant combinations. By contrast, arabidopsis single and double *er* family mutants show little to no change in SAM size, and only the *er erl1 erl2* triple mutant exhibits a dramatically broadened SAM[25]. These findings indicate that the maize ERECTA family likely exerts a stronger quantitative influence on meristem maintenance than in arabidopsis.

Importantly, disruption of *ZmER* genes directly increases kernel row number, linking ERECTA signaling to yield-related phenotypes. A similar function has been reported in rice, where *oser1* mutants exhibit increased spikelet number per panicle[31]. These findings together identify ERECTA as a conserved target for breeding or genome editing to improve inflorescence trait and yield potential. In addition, *Zmer1* and *Zmerl* single mutant, as well as all three double mutants, have an increased stomatal index, indicating that *ZmER* genes contribute to stomatal development (Supplementary Fig. 14). This is consistent with the well-characterized function of the ERECTA family in arabidopsis as negative regulators of stomatal development[57,58].

In Arabidopsis, four ER ligands EPFL1, EPFL2, EPFL4, and EPFL6 act redundantly to constrain meristem size and promote leaf initiation. The *epfl1,2,4,6* quadruple mutants have a compact shoot and inflorescence architecture similar to that of the *er* mutant[18]. In rice, the four OsER ligands OsEPFL6, OsEPFL7, OsEPFL8, and OsEPFL9 act synergistically to control rice panicle morphogenesis. Single *Osepfl6*, *Osepfl8*, and *Osepfl9* mutant already have enhanced spikelet number, and double mutants show an even greater increase[32,59]. These rice ligands also affect plant architecture in single and double mutant combinations, suggesting

overlapping roles in both vegetative and reproductive development. Similarly, our results indicate that maize EPFL signaling also contributes to inflorescence meristem regulation. We generated CRISPR-Cas9 knockout alleles in five *ZmEPFL* genes, including *ZmEPFL1-1*, *ZmEPFL1-2*, *ZmEPFL2-1*, *ZmEPFL4-2*, and *ZmEPFL9-1*, which are relatively highly expressed in IM and SAM. Among the mutant combinations identified, double and triple mutants that include *Zmepfl1-1* display enlarged, fasciated IMs and the quintuple mutants showed the most pronounced IM phenotype, while displaying no obvious changes in overall plant architecture. Together, these findings support a broadly conserved EPFL-ER signaling module that tunes meristem size and inflorescence architecture across species. Further analysis of additional allele combinations will be required to clarify the precise function of each *ZmEPFL* gene and the extent of functional redundancy among them. The maize genome encodes 16 EPF/EPFL type peptides[41]. Generating CRISPR-induced mutants for the remaining 11 peptide genes could provide a more comprehensive understanding of EPF/EPFL peptide effects on plant and ear architecture. Genetic manipulation of these EPF/EPFL peptides could further reveal their potential for yield improvement through optimizing different traits.

In arabidopsis, genetic evidence demonstrates that ER family and CLV3 signaling synergistically regulate meristem activities, with the *clv3 er1 erl1 erl2* quadruple mutant displaying the most dramatic meristem phenotype[27,29,60]. The ERECTA family signaling acts via EPFL ligands to laterally restrict *WUS* expression, complementing the CLV3-mediated central repression[28,29]. In *er1 erl1 erl2*, *WUS* expression is upregulated, and genetic analyses further revealed that WUS is

epistatic to ERECTA pathway, as *wus er1 erl1 erl2* exhibit a *wus*-like meristem and shoot phenotype[29]. In maize, we similarly found that *ZmER* genes negatively regulate *ZmWUS1* expression, with *ZmWUS1* being significantly upregulated in the IMs of the *Zmer1* single mutant as well as in *Zmer1;Zmer2* and *Zmer1;Zmerl* double mutants. Mutation of *Zmwus1* significantly, though not completely, suppressed the enlarged IM of *Zmer1*. However, *Zmwus1* cannot rescue the compact plant architecture of *Zmer1*. These findings suggest that the role of ZmER1 in IM regulation rely on ZmWUS1, whereas its function in vegetative architecture may involve additional or distinct components, highlighting both conserved and divergent aspects of ERECTA-WUS interactions across species. In addition, we found that ZmER1 genetically interacts with the key CLV component ZmCRN. While *Zmcrn* mutants didn't have obvious plant architecture defects, *Zmcrn* enhanced the plant architecture phenotypes of *Zmer1* in double mutants (Supplementary Fig. 15a). Furthermore, the *Zmer1;Zmcrn* double mutants further increased IM size compared with either single mutant, accompanied by a stronger upregulation of *ZmWUS1* expression (Supplementary Figs. 15b, c). These findings indicate that ZmERs and CLV pathways genetically interact in maize, reminiscent of the ER-CLV interactions reported in arabidopsis[29,61]. Together, our data suggest that ZmER1 perceives EPFL peptides, primarily EPFL1-1, with potential redundancy from other EPFLs (ZmEPFL1-2, ZmEPFL2-1, ZmEPFL4-2, and ZmEPFL9-1) to regulate IM development, possibly in part by constraining *ZmWUS1* expression.

RNA profiling for the IM of *Zmer1;Zmer2* and *Zmer1;Zmerl* mutants also identified genes involved in hormone biosynthesis and signaling, including upregulation of Aux/IAA and SAUR family auxin response genes[26,62], as well as differential expression of gibberellin GA oxidases genes[63], suggesting a role of auxin and GA in ZmER-mediated signaling (Fig. 5). In addition, we also detected concurrent upregulation of two jasmonate-pathway genes, *TS1*, encoding a lipoxygenase involved in JA biogenesis, and *SK1*, which encode a UDP glycosyltransferase that attenuates active jasmonates, which implies potential modulation of JA signaling in these mutants[64,65]. Nevertheless, no obvious sex determination phenotypes were observed, suggesting the JA homeostasis may be buffered in the developing florets. It would be interesting to explore how ZmER integrates with the CLV-WUS pathway and hormone signaling networks to regulate maize meristem development in the future. In summary, our findings provide valuable insights into how ZmER mediates meristem activity and coordinates plant and ear architecture in maize.

## Methods

### Plant materials and growth conditions
The *Zmer* mutants were generated using CRISPR/Cas9 gene editing of the HiII line at the Iowa State Plant Transformation Facility (Ames, IA) and the *Zmepfl* mutants were created through CRISPR/Cas9 gene editing of the KN5585 background at Weimi Biotechnology Co., Ltd. *Zmer1^{D295N}*, *Zmer1^{A369T}* and *Zmwus1^{w187*}* as isolated from an M3 screen of EMS-mutagenized B73 inbred maize (*Zea mays*) and was deposited in the Maize EMS Database as EMS4-13a82f, EMS4-089b7b, and EMS3-022073[45]. For phenotypic observations, maize plants were cultivated under natural growth conditions in fields located in Qingdao (Shandong Province, China) and Sanya (Hainan Province, China).

For laboratory experiments, maize plants were grown at under long-day conditions (14-h day/10-h night) in greenhouse.

### Phylogenetic tree construction
Sequences were aligned using the ClustalW alignment tool within MEGA X. Phylogenetic trees were generated using the Neighbor-Joining (NJ) method with 1,000 bootstrap replicates to assess the reliability of the tree topology. The resulting trees were visualized and annotated within MEGA X, with bootstrap values displayed to indicate the statistical confidence of each node. The phylogenetic trees were then exported in Newick format for further analysis and graphical refinement using the iTOL visualization software.

### Morphological measurements
Measurements of plant height, ear height, and leaf angle in maize were conducted after the silking stage, when the plant had reached maximum height. Plant height was measured as the vertical distance from the soil surface to the tip of the tassel, and ear height was the distance from the soil surface to the base of the primary ear. Leaf length was recorded as the distance from the leaf collar to the tip along the midrib, and leaf width was measured at the widest point of the blade. Leaf angle was determined using a protractor as the angle between the stalk and the midrib of the leaf blade. The upper leaf angle, leaf angle of the ear, and lower leaf angle refer to the leaf angle of the first leaf above the primary ear, the leaf at the primary ear, and the first leaf below the primary ear, respectively. Kernel row number (KRN) was measured on mature maize ears. Husk was removed, and the number of kernel rows was counted around the circumference at the midpoint of each ear. To prepare the leaves for stomatal peel, we applied clear nail polish to the same position on the abaxial epidermis of fully expanded leaves at the 10-leaf stage for different genotypes. After about 5-8 min, we removed the dried nail polish, placed it on pre-cleaned microscope slides, covered with one coverslip, and imaged using the inverted fluorescence microscope (NIKON TI-E) as described[66]. The stomatal index (SI) was calculated for each visual field. To measure internode cell size, the third internodes below the ear at the 12-leaf stage were sampled from WT, *Zmer1* single mutant, and *Zmer1;Zmer2* mutant, respectively. The sampled internodes were then stained with toluidine blue. The longitudinal section cells in the central region of the internode were observed using an inverted fluorescence microscope (NIKON TI-E). Three fields were observed for each internode, and ~10 cells per field were measured. The mean cell area from each field was used to represent the internode cell size for each genotype, and cell size was calculated using ImageJ software.

### Plasmid construction and plant transformation
The CRISPR/Cas9 system was used for gene editing of *ZmER1*, *ZmER2*, and *ZmERL*. We synthesized five guide RNAs (gRNAs) that specifically target three genes. These gRNAs were driven by different maize and rice U6 promoters and integrated into a pGW-Cas9 construct. Subsequently, the recombinant pGW-Cas9 construct carrying these gRNAs was transformed into the maize Hi-II genetic background for further investigation. A translational fusion of the coding sequence of ZmER1 and miniTurbo-YFP driven by the ZmER1 native promoter was created with the Multisite Gateway Pro kit (Invitrogen). We cloned a 10,552 bp ZmER1 fragment, including 6,953 bp genome sequence 2352 bp 5′-UTR and 1247 bp 3′-UTR into the binary vector pTF101 (pAM1006-RL). Genomic editing of *ZmER1* was identified by PCR amplification and Sanger sequencing of the target regions. The CRISPR/Cas9 system was used for gene editing of *ZmEPFL1-1*, *ZmEPFL1-2*, *ZmEPFL2-1*, *ZmEPFL4-1*, *ZmEPFL4-2* and ZmEPFL9-1, These gRNAs were driven by different maize and rice U6 promoters and integrated into the CPB-ZmUbiCas9 vector. The ZmEPF/EPFL CRISPR/Cas9 constructs used in this study were produced through seamless cloning with the 2X MultiF Seamless Assembly Mix and confirmed by sequencing. The relevant PCR primers are shown in Supplementary Data 1.

### Imaging and in situ hybridization
For scanning electron microscopy (SEM) imaging of the maize IM, ear primordia were fixed in 25% glutaraldehyde solution (prepared in 1× PBS), then sequentially dehydrated through a graded ethanol series. and subsequently processed for critical-point drying, followed by gold sputter coating. SEM imaging was conducted using Quanta 250 FEG system. For confocal microscopy, maize inflorescences were dissected, mounted in water on confocal petri dishes, and imaged using

the LSM900 Airyscan 2 confocal microscope (Zeiss, Germany). YFP signals were detected with a 514 nm laser for excitation and a 520–560 nm emission range for detection.

For histological analysis, two-week-old B73 SAM were freshly collected and fixed in 4% paraformaldehyde for 14 h at 4 °C. Samples were then dehydrated in a graded ethanol series, cleared with Histo-clear, and embedded in Paraplast. Sections of 8 μm thickness were cut using a Leica microtome and mounted on ProbeOn Plus slides. In situ hybridization experiments were performed as described[67] and primers used for probe design are listed in Supplementary Data 1.

## SAM measurements
Mutants and the WT siblings from the segregated F2 population were genotyped with primers listed in (Supplementary Data 1). Shoot apices were dissected from seedlings after 14 days of growth then dissected and fixed in FAA (10% formalin, 5% acetic acid, and 45% ethanol), dehydrated in an ethanol series, and cleared with methyl salicylate, mounted on glass slides, and imaged on a differential interference contrast (DIC) microscope (OLYMPUS BX53). The SAM size was measured using ImageJ.

## RNA extraction, RNA-Seq and RT-qPCR
The maize inflorescence meristems were dissected from 2-5 mm ear primordia and then subjected to total RNA extraction using the Direct-zol RNA extraction kit (Zymo Research). The RNA seq library was prepared and sequenced following the standard protocol on the Illumina platform at Yuanshen Biotechnology Co., Ltd (Shanghai, China), with each sample yielding 6 G of sequencing data. The raw sequencing reads were subjected to quality filtering using fastap (v0.20.1). Subsequently, the filtered reads were mapped to the B73 RefGen_v3 Reference using Hisat2 (v2.2.1) under default parameter settings. The read files were then processed using SAMtools (v1.7) for quality filtering (-q20) and conversion to BAM format. Gene-level expression values were quantified for all samples using StringTie (v2.1.4), with expression levels represented as fragments per kilobase of transcript per million mapped reads (FPKM). Differentially expressed genes were identified using an adjusted P-*value* (padj) <0.05 and $\log_2$ (FC) >1. Protein sequences of *Zea mays* were submitted to the EggNOG-MAPPER database (http://eggnog-mapper.embl.de/) for gene functional annotation. The resulting annotations were subsequently analyzed for GO enrichment using TBtools[68]. All GO terms with $P < 0.05$ were considered significant, and the top five most enriched biological process (BP) categories, together with representative GO terms from other categories, were selected for visualization.

cDNA was synthesized using the HiScript IV RT SuperMix for qPCR (+gDNA wiper) (Vazyme, R423-01) according to the manufacturer's manual. RT-qPCR was performed on a QuantStudio™ 5 Real-Time PCR System and used 2X Universal SYBR Green Fast qPCR Mix (ABclonal RK21203). *ZmUBI1* (*Zm00001d015327*) was used as the internal Reference gene to normalize expression data using the $2^{-\Delta\Delta CT}$ method. Primers are listed in Supplementary Data 1.

## Protein expression and purification
The coding sequences for the extracellular LRR domains of ZmER1 (residues 34–584, designed as ZmER1$^{LRR}$), and ZmEPFL2-1 (residues 46–128) were cloned into modified pGEX-4T-1-GST vectors and ZmEPFL1-2 (residues 46–143) and ZmEPFL9-1 (residues 79–123) were cloned into modified pET32a-His vectors. The plasmids were transformed into *E. coli* BL21 (DE3) cells. The expression of His-tagged and GST-tagged proteins were induced overnight at 16 °C by the addition of 0.5 mM isopropylthio-β-galactoside (IPTG) when cell density at OD600 reached 0.6 - 0.8. Proteins were purified by gravity flow with Ni Sepharose High Performance (cytiva) for His-tagged proteins and with BeyoGold™ GST-tag Purification Resin for GST-tagged proteins.

ZmEPFL1-1 (residues 89–157) and ZmEPFL4-2 (residues 63–113) proteins were synthesized and purified by GenScript Biotech Corporation.

## Microscale thermophoresis (MST) assay
The purified ZmER1$^{LRR}$-GST protein was fluorescently labeled using Monolith™ RED-NHS (MO-L001, Nano Temper, Germany). Subsequently, the labeled ZmER1$^{LRR}$-GST protein was incubated with peptides at a series of concentrations in PBS-T buffer (1× PBS with 0.05% Tween 20). ZmEPFL4-2, ZmEPFL1-1, and ZmFCP1 peptides were chemically synthesized. ZmEPFL2-1, ZmEPFL1-2, and ZmEPFL9-1 peptides were purified as described above in the "Protein expression and purification" section. Finally, the samples were loaded into silica capillaries (Monolith™ NT.115 standard Treated capillary, MO-K002) and measured using the Monolith NT.115 (Nano Temper Technologies). The data were analyzed using MO. Affinity Analysis software.

## Statistical analysis
Statistical analysis was performed using the Student's *t*-test or one-way ANOVA ($P < 0.05$; Tukey's multiple comparisons test). The data are shown as mean standard deviation (SD). All analyses were performed using GraphPad Prism 9.5 software and were shown in the graphs or source data.

## Reporting summary
Further information on research design is available in the Nature Portfolio Reporting Summary linked to this article.

## Data availability
The RNA-seq data generated in this study have been deposited in the National Center for Biotechnology Information [https://www.ncbi.nlm.nih.gov/sra/PRJNA1263037]. Source data are provided with this paper.

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

## Acknowledgements

This work was supported by the National Natural Science Foundation of China (U22A20474 to F.X., U22A20460 to X.H., and 32101741 to F.X.), Natural Science Foundation of Shandong Province (ZR2023JQ012 to F.X.) and the Taishan Scholar Program of Shandong Provincial Government to F.X., and US NSF (Grant IOS-2131631 and IOS-2129189 to D.J.). We thank Prof. Xiaoduo Lu at Qilu Normal University and Prof. Chunyi Zhang at Biotechnology Research Institute, Chinese Academy of Agricultural Sciences, for providing the EMS alleles of *Zmer1 and Zmwus1*. Thanks to Sen Wang, Haiyan Yu, Xiaomin Zhao and Yuyu Guo from Core Facilities for Life and Environmental Sciences at the SKLMT (State Key Laboratory of Microbial Technology, Shandong University) for the assistance provided in scanning electron microscopy and laser scanning confocal microscopy. We also thank Feng Zhang (Core Facility and Service Platform, School of Life Sciences, Shandong University) for technical support with the SpectraMax® i3x system.

## Author contributions

F.X., D.J., and X.L. conceived this project. X.L., J.W., J.L., L.K., M.W., J.M., and X.H. collected tissue samples and performed experiments. J.W. and Z.H. performed transcriptome analysis. X.L. prepared the figures and, together with F.X. and D.J., wrote the paper. Z.Z., F.Y., M.B., L.W., and Z.D. revised the manuscript.

## Competing interests

The authors declare no competing interests.
