## [Transparent Peer Review file · Nature Communications]

ERECTA genes and their ligands regulate shoot and inflorescence architecture in maize

Corresponding Author: Professor Fang Xu

Version 0:

Reviewer comments:

Reviewer #1

(Remarks to the Author)

This manuscript describes the role of ERECTA signaling in maize. The authors demonstrate that three ERECTA receptors function redundantly to regulate stem height, the size of both vegetative and inflorescence shoot apical meristems (SAM), leaf angle and size, kernel row number, and ear size. Among these, ZmER1 appears to play a primary role in the regulation of several key developmental processes. The data links multiple EPFL ligands to the control of inflorescence SAM size. In addition, the authors show that, as in Arabidopsis, ERECTA signaling in maize regulates the expression of the transcription factor WUS. This work highlights the conserved role of ERECTA signaling across eudicots and monocots, while also providing specific insights into its function in a major crop species. Finally, the analysis of weak ERECTA alleles suggests that manipulating this pathway could enhance yield-related traits in maize.

In addition, the authors attempt to extend the study by proposing that ZmER functions in complex with the kinase ZmCRN; however, the current data do not sufficiently support this claim. A substantial amount of additional work would be required to validate this interaction, and I recommend removing this section and pursuing it as a separate study when more evidence is available.

The greatest strength of this manuscript lies in its agricultural relevance, given the economic and global importance of maize as a crop.

Major issues requiring additional data:

1. Line 192: The authors state that “these five ZmEPFL peptides redundantly regulate IM development.” However, this conclusion is not fully supported by the data presented. While it is evident that the increased inflorescence phenotype depends on ZmEPFL2-1, the contributions of the other ZmEPFLs remain unclear. For instance, the phenotype could result from redundancy between ZmEPFL2-1 and ZmEPFL1-1, while the remaining three peptides may not be involved in inflorescence meristem development. To substantiate the claim of redundancy among all five peptides, it would be necessary to analyze all possible combinations of quadruple mutants and show that these either lack a phenotype or exhibit milder phenotypes. Alternatively, the authors could revise their wording to avoid overstating their conclusions.
2. The finding that ZmER1 interacts with ZmCRN is intriguing and potentially important, but additional data are needed to support this conclusion convincingly. Previous literature has clearly established CRN as a component of the CLV signaling pathway. The split-luciferase and co-immunoprecipitation assays would be more compelling if they included FEA2 (the maize ortholog of CLV2) as a positive control, and compared the strength of ZmCRN’s interactions with both ZmER and FEA2. In Arabidopsis, genetic evidence demonstrates strong interactions between the ERECTA and CLV pathways, as both contribute to repression of WUS expression. The phenotype of the Zmer1 Zmcrn double mutant could reflect synergy between the CLV and ZmER pathways. If ZmCRN functions exclusively within the ZmER pathway, then the phenotype of Zmer triple and Zmer triple Zmcrn mutants should be identical. An alternative approach would be to compare the phenotypic effects of Zmer mutants combined with Zmcrn to those combined with known CLV pathway mutants. If adding Zmcrn to Zmer mutants yields a phenotype resembling that of Zmer + CLV mutants, this would suggest ZmCRN primarily functions in the CLV pathway. Conversely, if the Zmer1 Zmcrn phenotype simply enhances the Zmer phenotype, it would support a more direct role for ZmCRN in the ERECTA pathway.
3. Line 193 The statement “Quadruple mutants also had slightly enlarged meristems” requires supporting statistical analysis.
4. Line 250, The statement “the enlarged inflorescence meristem was significantly suppressed by Zmwus1” requires supporting statistical analysis.

Minor issues requiring clarification:

1. Line 70. Citations 23 and 24 do not support the statement. Instead, reference 31 is an appropriate citation.
2. Line 84. Add the following citation "Rapid customization of Solanaceae fruit crops for urban agriculture" by Kwon et al 2020.
3. Fig. S1B and S10 indicate that ZmER1/2 have 8 LRR repeats and ZmERL has 10. This is incorrect. ER/ERL proteins always have 20 consecutive LRR repeats in eudicots and monocots, with no protein sequences in between them.
4. Please describe how you measured leaf angle and ear height in the methods, including how old the plants were when used for measurements.
5. Legend of Fig. S4. Change "scalebar, 20cm (a,d)" to "scalebar, 20cm (a,c)"
6. The legend of Fig. 1 says there is a 100 μ m scale bar in panel f, but no scale bar is shown in that panel."
7. Fig. S8. Please include a phylogenetic tree that incorporates Arabidopsis EPF/EPFL family members. This would clarify how the ZmEPF/EPFL gene names correspond to their Arabidopsis counterparts. For example, it is unclear whether the current naming of ZmEPFL9 and ZmEPFL1 is appropriate, given that Arabidopsis EPFL1 and EPFL2 are more closely related to each other than to other EPF/EPFLs. Similarly, the placement of ZmEPFL4-1 as more closely related to ZmEPFL5 than to ZmEPFL4-2 raises questions about whether the names ZmEPFL5 and ZmEPFL4-1 should be switched for consistency.
8. Line 181; Authors state, "we selected six EPFL genes that were highly expressed in the SAM and IM, based on the available RNA seq". Fig. 8. Based on that Figure ZmEPFL9-1 is not the most expressed gene. Either ZmEPFL4-2, ZmEPFL8, or ZmEPFL2-2 would be a better choice. Why was ZmEPFL9-1 selected?
9. The top labels of Fig.5a are incorrect. ZmER2-KD-AD and ZmERL-KD-AD are labeled as ZmER1-KD-AD.
10. Describe the logic of using ZmFEA3 and ZmBIP1 as controls for a split-luciferase assay in the methods. Why these two genes?
11. Use of gene symbols and full names in Supplemental data #2 and #3 (columns I and J) is random. For example, ZmWUS1, which is actively discussed in the results, is not labeled, while some other genes that are never discussed are labeled. Why? I recommend labeling as many genes as possible.
12. Fig. 6. Does functional enrichment analysis of RNA-seq data identify any biological themes or functions?

Reviewer #2

(Remarks to the Author)

Remarks to the author

This is a very thorough characterization of the function of the ERECTA pathway in maize including the identification of three receptors and five ligands and role in vegetative and reproductive development. My only concern was with the writing of the discussion which basically repeated the results. Please compare and contrast the results to the previously published work in Arabidopsis and rice.

Information that needs to be added

Figure 1 – it is difficult to distinguish the expression of ZmER1 from the Turbo-YFP pictures. Can you add an RNA in situ hybridization so that the expression can be compared to the other two ER genes.

Since ERECTA is well known to affect stomatal development in Arabidopsis, where there are any stomatal defects in the ZmER mutants?

Figure 7 – please explain how kernel row number was measured. This information was not in the materials and methods.

Suppl Fig 4 – what was the internode length of the mutants? Where the short internodes due to smaller cells?

Suppl Fig 8 – add at least the Arabidopsis and rice genes to the ZmEPFL phylogeny. The relationships are impossible to determine otherwise.

Minor concerns

Figure 3 – there are two typos in parts d and e. "quintuple"

Where off target effects checked in the EPF and EPFL CRISPR lines?

Figure 6 – since TS1 and SK1 were differentially expressed – where there sex determination effects in the ZmER double mutants? Do you have an explanation for why these genes are increased in expression?

Is ref1 the correct ref? Include original references to td1, fea2 etc?

Suppl Fig 1 ER phylogeny – since the rice genes are published please add their names beside the gene ID's as was done for Arabidopsis and Maize

Suppl Fig 5 and Fig 7 RPM of maize ER genes – add references to where this data came from

Suppl Fig 6 –The WT control here included the transgene. Since the wt without transgene was not shown, where there any additional phenotypes from adding a second copy of ZmER?

Reviewer #3

(Remarks to the Author)

This study from the groups of Drs. Jackson and Xu investigates the functions of the conserved ERECTA (ER) family members and their ligands, EPIDERMAL PATTERNING FACTOR-Like (EPFL) peptides, in regulating shoot meristem size, plant architecture, and ear development in maize. By integrating CRISPR-mediated gene editing, molecular genetic analysis, biochemical assays, and transcriptomic studies, the authors show that a member of the ER family, directly interacts

with ZmCRN to restrict ZmWUS1 expression, thereby controlling meristem development. The majority of the datasets presented are solid and convincing, and the ZmER1-mediated genetic pathways proposed in this work offer valuable insights into the conservation and divergence of key regulatory components and their family members in controlling shoot meristem development in flowering plants.

The reviewer only has a few comments and suggestions that may help strengthen the manuscript:

1. Proposed Genetic Pathway of ZmER1–ZmWUS1

The authors demonstrated that ZmWUS1 is upregulated in Zmer1;Zmer2 and Zmer1;Zmerl mutants based on both RNA-seq and qPCR data, and further showed that the enlarged meristem phenotype of Zmer1 is dependent on ZmWUS1. These findings are interesting, but a few points need to be addressed: 1) In Fig. 6f, the authors showed that Zmwus1 suppressed the enlarged inflorescence meristem (IM) phenotype of Zmer1. To make this result more convincing, please indicate how many independent biological replicates were included in the analysis and provide quantitative measurements of meristem size across the four genotypes. 2). The plant growth images in Fig. 6e suggest that Zmwus1 moderately rescues the dwarf phenotype of the Zmer1 mutant. Is this because plant architecture and meristem size can be uncoupled, and ZmER1 specifically regulates meristem development through ZmWUS1? A clearer interpretation of this observation would be helpful. Additionally, is ZmWUS1 upregulation specific to the meristem, or is it also elevated in non-meristematic tissues? RNA in situ hybridization of ZmWUS1 in Zmer1 single or double mutants would help address this question. 3) Following up on this point: are there any misregulated genes in Zmer1 mutants that are dependent on functional ZmWUS1?

2. Characterization of weak Zmer1 alleles

The reviewer appreciates the authors' efforts in identifying and characterizing weak Zmer1 alleles as potential resources for yield improvement. However, there appears to be a discrepancy between the two weak alleles analyzed. The Zmer1A369T allele shows a statistically significant increase in kernel row number (KRN) compared to WT, whereas the difference between WT and Zmer1D295N is not statistically supported (Fig. 7g). Additionally, several other traits appear unchanged in both alleles compared to WT (Fig. 7h-j). Given these observations, the interpretation regarding the yield-enhancing potential of Zmer1 weak alleles should be made more cautiously.

3. Proposed ZmER1–ZmCRN Interaction

The authors present compelling evidence for both physical and genetic interactions between ZmER1 and ZmCRN, providing new insight into crosstalk between the ERECTA and CLAVATA pathways in meristem regulation. However, a key piece of evidence is missing to fully support the proposed working model (Supplementary Fig. 11): What is the functional output of the ZmER1–ZmCRN interaction? Specifically, is ZmWUS1 expression further upregulated in the Zmer1; Zmcrn double mutant compared to WT or either single mutant? This analysis would help solidify the proposed model.

Version 1:

Reviewer comments:

Reviewer #1

(Remarks to the Author)

Previously, I had two major criticisms. The first concerned the authors' claim that they had identified ZmEPFL genes regulating inflorescence meristem (IM) development, which was not sufficiently supported by the data. The revised manuscript now includes convincing evidence linking ZmEPFL1-1 to the regulation of IM size. However, the contributions of other EPFLs remain unclear. Based on the available data, it appears that among ZmEPFL1-2, ZmEPFL2-1, ZmEPFL4-2, and ZmEPFL9-1, some peptides influence IM size in a partially redundant manner. While the authors have modified their conclusion in the text to acknowledge this point, Figure S11 still depicts all other EPFLs as regulators of WUS expression. Please revise this figure accordingly. Additionally, it would be valuable to include an image of a single Zmepfl1-1 mutant along with quantitative measurements of its IM size. This would help determine whether the phenotypes observed in Zmepfl1-1 Zmepfl1-2 and Zmepfl1-1 Zmepfl9-1 mutants are primarily due to the Zmepfl1-1 mutation, providing definitive evidence that ZmEPFL1-1 regulates IM size.

My second criticism concerns the inclusion of ZmCRN as part of the ER signaling pathway. The authors still have not provided sufficient data to support this conclusion. In the revised version, they added one additional experiment showing that FEA2 interacts with CRN with much higher affinity compared to ZmER1. The weak interaction observed between CRN and ER1 could simply result from protein overexpression in tobacco leaves. The presented evidence is not conclusive that ZmCRN functions within the ER signaling pathway, and I continue to strongly recommend removing this claim from the manuscript. While I agree that "it is not uncommon for one protein to participate in different protein complexes," the current data do not convincingly demonstrate such a role for ZmCRN with ERs. All protein interaction assays (yeast two-hybrid and split-luciferase) were conducted under overexpression conditions, in which many non-specific interactions can occur. Furthermore, the reported genetic interactions are questionable, as a synergistic relationship would be expected between the FEA2 pathway and ERs. I do not understand the authors' insistence on maintaining this unsupported conclusion in an otherwise strong manuscript. Although the manuscript text was slightly revised to temper the conclusion, the changes are not sufficient, and the model in Figure S11 still depicts ZmCRN.

All my remaining criticisms and suggestions have been addressed satisfactorily.

Reviewer #3

(Remarks to the Author)

In the revised manuscript, the authors have included new experimental data and have revised the descriptions and discussion. I appreciate the authors' efforts and careful revisions, and I believe that the revised manuscript has been significantly improved, addressing all of my previous comments. I have no further comments or concerns.

We sincerely appreciate the constructive comments and suggestions from the three reviewers and the editor. In the revised manuscript, we analyzed additional *Zmepfl* mutant combinations to show their roles in meristem development and refined the wording throughout to avoid overstatement. We include FEA2 as the control in the ZmER and ZmCRN interaction assays, indicating that ZmCRN does not function exclusively within ZmER signaling pathway. We further checked the *ZmWUS1* expression in *Zmcrn*, *Zmer1* and the *Zmcrn;Zmer1* double mutant to support their genetic interaction.

We quantified plant height, ear height, and IM size in WT, *Zmer1*, *Zmwus1* and *Zmer1;Zmwus1*, supporting that the role of ZmER1 in inflorescence meristem development largely relies on ZmWUS1. We performed *in situ* hybridization for *ZmER1* to enable direct comparison with *ZmER2* and *ZmERL* expression pattern. We examined stomatal epidermal patterning in *Zmer* single and double mutants and found that ZmER proteins contribute to stomatal development. We measured internode length in the *Zmer1* single and double mutant and showed that the reduction correlated with smaller cell size. We reconstructed the ZmEPFL phylogenetic tree and carried out the GO enrichment analysis of the RNA seq data. We also expand the discussion by comparing our results with findings in arabidopsis and rice, and highlighted the new insights from maize.

We believe we have addressed all the reviewer's questions and revised the manuscript accordingly. Point-by-point responses to the reviewer's comments are in blue below the original statements, and corresponding changes in the revised manuscript are also highlighted in blue.

Reviewer #1 (Remarks to the Author):

This manuscript describes the role of ERECTA signaling in maize. The authors demonstrate that three ERECTA receptors function redundantly to regulate stem height, the size of both vegetative and inflorescence shoot apical meristems (SAM), leaf angle and size, kernel row number, and ear size. Among these, ZmER1 appears to play a primary role in the regulation of several key developmental processes. The data links multiple EPFL ligands to the control of inflorescence SAM size. In addition, the authors show that, as in Arabidopsis, ERECTA signaling in maize regulates the expression of the transcription factor WUS. This work highlights the conserved role of ERECTA signaling across eudicots and monocots, while also providing specific insights into its function in a major crop species. Finally, the analysis of weak ERECTA alleles suggests that manipulating this pathway could enhance yield-related traits in maize.

In addition, the authors attempt to extend the study by proposing that ZmER functions in complex with the kinase ZmCRN; however, the current data do not sufficiently support this claim. A substantial amount of additional work would be required to validate this interaction, and I recommend removing this section and pursuing it as a separate study when more evidence is available.

Thank you for the suggestion, we have added FEA2 as the positive control and compared the interaction strength between ZmER-ZmCRN and FEA2-ZmCRN. Please see response to the 2nd major point.

The greatest strength of this manuscript lies in its agricultural relevance, given the economic and global importance of maize as a crop.

Thank you for the comments and suggestions.

Major issues requiring additional data:

1. Line 192: The authors state that “these five ZmEPFL peptides redundantly regulate IM development.” However, this conclusion is not fully supported by the data presented. While it is evident that the increased inflorescence phenotype depends on ZmEPFL2-1, the contributions of the other ZmEPFLs remain unclear. For instance, the phenotype could result from redundancy between ZmEPFL2-1 and ZmEPFL1-1, while the remaining three peptides may not be involved in inflorescence meristem development. To substantiate the claim of redundancy among all five peptides, it would be necessary to analyze all possible combinations of quadruple mutants and show that these either lack a phenotype or exhibit milder phenotypes. Alternatively, the authors could revise their wording to avoid overstating their conclusions.

Thank you very much for the suggestion. We agree that isolating additional combinations is important for testing their genetic relationship. As suggested, we tried to isolate more mutant combinations in the different editing lines. During this process, we discovered that some *Zmepfl1-1^{cr1}* alleles carries an additional 1-bp insertion in the 2nd exon, in addition to the original 4bp deletion in the 1st exon. After splicing, this compound allele is predicated to alter only six amino acids without causing a frameshift, and therefore should not be considered null. Accordingly, we have removed mutant combinations containing this additional 1-bp insertion in the revised manuscript. This insertion site is located relatively far from the gRNA target (separated by an 132bp intron) and was detected in some, but not all, lines. This unintended mutation was not anticipated and it was overlooked in our initial analysis. We apologize for this oversight and are grateful to the reviewer for promoting us to make additional mutant analysis.

We have carefully re-sequenced and analyzed all the five EPFL gene in both the previously described mutants and the newly isolated mutant combinations to ensure that all the lines used carry frameshift, loss-of-function alleles. In addition, the ZmEPFL5 has been renamed as ZmEPFL4-2 as suggested based on the phylogenetic tree in the revised manuscript (see response to the 7th minor point). The genotypes and phenotypes of re-analyzed lines are provided in the revised **Fig. 3** as shown below:

From these analyses, we found that two double mutants *Zmepfl1-1;Zmepfl1-2* and *Zmepfl1-1;Zmepfl9-1*, as well as two triple mutants *Zmepfl1-1;Zmepfl1-2;Zmepfl4-2* and *Zmepfl1-1;Zmepfl1-2; Zmepfl9-1* exhibit noticeable meristem fasciation (updated **Fig. 3**, shown below). By contrast, the single mutants *Zmepfl1-2*, *Zmepfl2-1*, and *Zmepfl4-2*; the double mutants *Zmepfl1-2;Zmepfl4-2* and *Zmepfl2-1;Zmepfl4-1*, and triple mutants *Zmepfl1-1;Zmepfl1-2;Zmepfl4-2* and *Zmepfl1-2; Zmepfl4-2;Zmepfl9-1*, showed normal IM development (updated **Fig. 3**, shown below). These results suggest a predominantly role of *ZmEPFL1-1* in regulating meristem size, since combinations including *Zmepfl1-1* displayed clear IM fasciation, whereas other double and triple combinations did not. We did not isolated additional quadruple mutants, and the previous described *Zmepfl1-1;Zmepfl1-2;Zmepfl4-2;Zmepfl9-1* line is no longer considered a true quadruple mutant, as the *Zmepfl-1^{cr1}* allele in this line was found to contain the additional 1bp insertion and should not be considered null. Consequently, this quadruple mutant has been removed from the revised manuscript. The previous presented quintuple mutants remain valid, as sequencing confirmed that *Zmepfl1-1^{cr1}* allele in that line does not contain the extra 1-bp insertion. In addition, we identified a new quintuple mutant allele in the new round of analysis, which also showed massively overproliferated IM (updated **Fig. 3**, shown below). Notably, the quintuple mutant exhibited the most severe fasciation of the inflorescence meristem among all the mutant combinations examined.

We agree with the reviewer that fully resolving the functions and genetic relationships of these *EPFL* genes would require obtaining all possible mutant combinations. However, we currently were not able to obtain all possible mutant combinations, as generating every combination across five loci is technically challenging due to the low segregation ratios. We hope our current mutant characterization provide supporting evidence for the involvement of these *EPFL* genes in maize meristem development. In the revised manuscript, we have carefully rephrased the text to avoid overstating their functions or genetic relationships. Further work to comprehensively dissect these five *EPFL* genes and potentially other members of this family can be pursued in future studies to clarify their roles in regulating shoot and inflorescence architecture.

We have updated the **Fig. 3** as shown below:

Fig. 3 *ZmEPFLs* regulate the development of inflorescence meristems. **a** CRISPR/Cas9 editing generated different distinct frameshift alleles in five *EPFL* genes. **b** Representative images of overall plant architecture in the indicated mutant combinations. **c** Images of young ear primordia showing the IM development in different mutant combinations. Scale bars, 20 cm (**b**), 500 μ m (**c**).

The manuscript text has been rephrased accordingly to ensure that our conclusions are appropriately stated, as follows:

“...With the mutant combinations obtained, the single mutants *Zmepfl1-2*, *Zmepfl2-1*, and *Zmepfl4-2*; the double mutant *Zmepfl1-2;Zmepfl4-2* and *Zmepfl2-1;Zmepfl4-2*; and the triple mutant *Zmepfl1-1;Zmepfl1-2;Zmepfl4-2* and *Zmepfl1-2;Zmepfl4-2;Zmepfl9-1*, have normal IM development (Fig. 3c). By contrast, the double mutants *Zmepfl1-1;Zmepfl1-2* and *Zmepfl1-1;Zmepfl9-1*, as well as the triple mutants *Zmepfl1-1;Zmepfl1-2;Zmepfl4-2* and *Zmepfl1-1;Zmepfl1-2;Zmepfl9-1*, exhibit noticeable IM fasciation (Fig. 3c). These results suggest that *ZmEPFL1-1* plays a relatively predominant role in regulating meristem size, as combinations including *Zmepfl1-1* displayed fasciated IMs, whereas other double and triple combinations did not. Notably, the quintuple mutant displayed the most pronounced IM fasciation among all genotypes examined, which may reflect partial functional redundancy among these *ZmEPFL* peptides (Fig. 3). No mutant combinations including the quintuple mutant showed obvious changes in overall plant architecture (Fig. 3), implying that the *ZmEPFL* genes may primarily influence inflorescence meristem development. Nonetheless, further work involving all possible mutant combinations will be required to fully clarify the specific functions and genetic relationships of individual *ZmEPFL* genes in plant and meristem development...”

2. The finding that ZmER1 interacts with ZmCRN is intriguing and potentially important, but additional data are needed to support this conclusion convincingly. Previous literature has clearly established CRN as a component of the CLV signaling pathway. The split-luciferase and co-immunoprecipitation assays would be more compelling if they included FEA2 (the maize ortholog of CLV2) as a positive control, and compared the strength of ZmCRN's interactions with both ZmER and FEA2. In Arabidopsis, genetic evidence demonstrates strong interactions between the ERECTA and CLV pathways, as both contribute to repression of WUS expression. The phenotype of the *Zmer1 Zmcrn* double mutant could reflect synergy between the CLV and ZmER pathways. If ZmCRN functions exclusively within the ZmER pathway, then the phenotype of *Zmer* triple and *Zmer* triple *Zmcrn* mutants should be identical. An alternative approach would be to compare the phenotypic effects of *Zmer* mutants combined with *Zmcrn* to those combined with known CLV pathway mutants. If adding *Zmcrn* to *Zmer* mutants yields a phenotype resembling that of *Zmer* + CLV mutants, this would suggest ZmCRN primarily functions in the CLV pathway. Conversely, if the *Zmer1 Zmcrn* phenotype simply enhances the *Zmer* phenotype, it would support a more direct role for ZmCRN in the ERECTA pathway.

Thank you for the suggestion. We have now included FEA2 as a positive control in both the split luciferase and co-immunoprecipitation assays, which further support the interaction between ZmCRN and ZmER proteins. Both experiments showed that, although the interaction between

ZmER proteins and ZmCRN is generally not stronger than that between FEA2 and ZmCRN (newly added **Supplementary Fig.10a** and **10b**), the difference was not that pronounced, as both interactions can be detected under the same experimental conditions. These results suggest that ZmCRN may not function exclusively within the ZmER pathway.

The newly added **Supplementary Fig. 10** is shown below:

Supplementary Fig.10. ZmERs interact with ZmCRN in split luciferase and Co-immunoprecipitation assays, with the FEA2 serve as the positive control. a Split luciferase assay showed that ZmCRN interacts with both ZmERs and ZmFEA2. The interaction between ZmCRN and FEA2 appeared stronger than that between ZmCRN and ZmERs. **b** Co-immunoprecipitation assays were performed to compare the interaction strengths of ZmCRN with FEA2 and ZmER1. Less ZmER1 was co-immunoprecipitated compared with FEA2, with similar amount of ZmCRN being pulled down, suggesting that the ZmCRN-FEA2 interaction is stronger than the ZmCRN-ZmER1 interaction. The arrow indicates the position of the ZmCRN-GFP band, the asterisk marks the ZmER1-Myc band, and the triangle denotes the ZmFEA2-Myc band.

It is not uncommon for one protein to participate in different protein complexes. For example, BAK1 (BRASSINOSTEROID INSENSITIVE 1-associated receptor kinase 1), act in different signaling complexes in development and defense (Chinchilla D et al., 2009, Trends Plant Sci; Huang WRH & Joosten MHAI, 2025, Trends Plant Sci). Thus, it is plausible that ZmCRN can physically function in both CLV and ZmER pathways. In addition, although the *Zmcrn* single mutant did not show obvious plant architecture defects, it significantly enhanced the compactness of *ZmER1* (Fig. 5d). This enhancement may reflect their physical interaction, as reported in a recent study (Chen W et al., 2022, Science), which showed that while the KRN2 (Kernel Row Number 2) interactor DUF1644 alone exhibits no clear phenotype, mutation of DUF1644 markedly

enhances the increased IM size and kernel row number of *krn2*. Therefore, we prefer to retain the interaction data, as it may be relevant.

With the newly added data, the manuscript was revised as follows:

“...We further tested the interaction between ZmCRN and full-length ZmERs proteins using a split-luciferase assay, which confirmed that all three ZmERs can interact with ZmCRN (Fig. 5b). The interaction between ZmCRN and ZmER1 appeared stronger than those with ZmER2 or ZmERL, and, overall, interactions between the ZmER family and ZmCRN were not stronger than that between ZmCRN and the CLV receptor FEA2 in this assay (Supplementary Fig.10a). The ZmER1-ZmCRN interaction was further validated by a co-immunoprecipitation experiment (Fig. 5c), although it appeared weaker than the interaction between FEA2 and ZmCRN (Supplementary Fig.10b)...”

We also agree with the reviewer that the *Zmer1;Zmcrn* double mutant phenotype could reflect synergy or additive effects between the CLV and ZmER pathways. It is possible that the observed double mutant phenotype results from a combination of genetic additivity and protein-level interaction. Accordingly, we have carefully revised the manuscript to avoid overemphasizing the protein interaction and to discuss both genetic and physical interaction in the Discussion section as follows:

“...In maize, we found that ZmER1 genetically interacts with the key CLV component ZmCRN. The *Zmer1;Zmcrn* double mutant displayed a further increase in IM size compared with either single mutant, accompanied by stronger upregulation of *WUS* expression (Supplementary Fig 16). These findings indicate that ZmERs and CLV pathways genetically interact in maize, reminiscent of the ER-CLV interactions reported in arabidopsis^{29,62}. Interestingly, we also detected a physical interaction between ZmER1 and ZmCRN, as revealed by yeast two-hybrid, split-luciferase and co-immunoprecipitation assays, suggesting the possibility of direct crosstalk between ER and CLV pathway. Given that ZmCRN is known to function together with the CLV receptor FEA2, its interaction with ZmER1 suggests that ZmCRN may play dual roles in both CLV and ER signaling...”

Since our data indicate that ZmCRN does not function exclusively within ZmER pathway, and generating the *Zmcrn;Zmer1;Zmer2;Zmerl* quadruple mutants or double mutant between *Zmer1* and other *clv* mutants would require at least one more year of work, we were unable to provide these genetic analysis in the current study.

3. Line 193 The statement “Quadruple mutants also had slightly enlarged meristems” requires supporting statistical analysis.

Re-sequencing of the previously described quadruple line revealed that the *Zmepfl1-1^{cr1}* alleles carries an additional 1-bp insertion in the second exon, in addition to the original 4bp deletion in the first exon. After splicing, this compound allele is predicated to alter only six amino acids without causing a frameshift (see also the response to the first major point), and therefore cannot be considered a null allele. Consequently, the previously presented “quadruple mutant” is not true quadruple. To avoid confusion, we have omitted the data and the corresponding statement from the revised manuscript.

4. Line 250, The statement “the enlarged inflorescence meristem was significantly suppressed by *Zmwus1*” requires supporting statistical analysis.

As suggested, we quantified the inflorescence meristem (IM) size of wild-type, *Zmwus1*, *Zmer1*, and the *Zmer1*;*Zmwus1* double mutant within segregating populations. The results show that the *Zmer1* IM is significantly increased compared to the WT and this enlargement is significantly suppressed by *Zmwus1*. The new data was added in the revised **Fig.6i** as follows:

Fig. 6i. Quantification of IM size of WT, *Zmer1*, *Zmwus1* and *Zmer1*;*Zmwus1* shows that *Zmer1* significantly suppresses the enlarged IM of *Zmer1*. Although IM size in the double mutant remained bigger than the WT, the reduction was substantial. Data were presented as the means \pm s.d., *** $P < 0.001$, * $P < 0.05$, from an unpaired two-tailed Student’s *t*-test.

And the manuscript text was revised accordingly as follows:

“...In contrast, *Zmwus1* mutant significantly suppressed the enlarged IM of *Zmer1*, with the IM size of *Zmer1*;*Zmwus1* double mutant being significantly smaller than that of *Zmer1*(Fig.6h-i)....”

Minor issues requiring clarification:

1. Line 70. Citations 23 and 24 do not support the statement. Instead, reference 31 is an appropriate citation.

Thank you for pointing this out. As suggested, we have replaced the citations 23 and 24 with the proper reference in the revised manuscript.

2. Line 84. Add the following citation "Rapid customization of Solanaceae fruit crops for urban agriculture" by Kwon et al 2020.

As you suggested, we added the reference of Kwon et al 2020 here.

3. Fig. S1B and S10 indicate that ZmER1/2 have 8 LRR repeats and ZmERL has 10. This is incorrect. ER/ERL proteins always have 20 consecutive LRR repeats in eudicots and monocots, with no protein sequences in between them.

Thank you for pointing out this mistake. The original protein structure diagram, based on the online SMART prediction, failed to identify all the LRR repeats. We have now reanalyzed the protein sequences using the LRR Search Tool and Database and carefully aligned the maize ERECTA proteins with their arabidopsis homologs to obtain a more accurate structure prediction. Our analysis confirms that the maize ZmER/ERL proteins contain 20 consecutive LRR repeats. Accordingly, we have updated **Fig. S1b** with the revised protein structure diagram shown below:

Supplementary Fig. 1b. Schematic representation of the maize ER protein structure with CRISPR/Cas9 gRNA target sites indicated by inverted triangles.

4. Please describe how you measured leaf angle and ear height in the methods, including how old the plants were when used for measurements.

As you suggested, we added the information in the method section as follows:

"... Measurements of plant height, ear height, and leaf angle in maize were conducted after the silking stage, when plant had reached maximum height. Plant height was measured as the vertical

distance from the soil surface to the tip of the tassel, and ear height was the distance from the soil surface to the base of the primary ear. Leaf length was recorded as the distance from the leaf collar to the tip along the midrib, and leaf width was measured at the widest point of the blade. Leaf angle was determined using a protractor as the angle between the stalk and the midrib of the leaf blade. The upper leaf angle, leaf angle of the ear and lower leaf angle refer to the leaf angle of the first leaf above the primary ear, the leaf at the primary ear, and the first leaf below the primary ear, respectively..."

5. Legend of Fig. S4. Change "scalebar, 20cm (a,d)" to "scalebar, 20cm (a,c)"

Thank you for catching the mistake. The error in the legend of Fig. S4 has been corrected in the revised manuscript.

6. The legend of Fig. 1 says there is a 100 μ m scale bar in panel f, but no scale bar is shown in that panel."

Thank you for pointing this out. The 100 μ m scale bar should have been indicated for panel g, not panel f. We have now corrected this error in the revised figure legend.

7. Fig. S8. Please include a phylogenetic tree that incorporates Arabidopsis EPF/EPFL family members. This would clarify how the ZmEPF/EPFL gene names correspond to their Arabidopsis counterparts. For example, it is unclear whether the current naming of ZmEPFL9 and ZmEPFL1 is appropriate, given that Arabidopsis EPFL1 and EPFL2 are more closely related to each other than to other EPF/EPFLs. Similarly, the placement of ZmEPFL4-1 as more closely related to ZmEPFL5 than to ZmEPFL4-2 raises questions about whether the names ZmEPFL5 and ZmEPFL4-1 should be switched for consistency.

Thank you for your suggestion. We have now incorporated the EPF/EPFL homologs from Arabidopsis and rice to generate a more comprehensive phylogenetic tree, which clarifies how the ZmEPF/EPFL gene names correspond to their counterparts in Arabidopsis and rice. We believe the naming of ZmEPFL9 and ZmEPFL1 is appropriate as they cluster with AtEPFL9 and AtEPFL1, respectively. As you pointed out, the previous naming of ZmEPFL5 and ZmEPFL4-2 was not appropriate. We have now switched the names of ZmEPFL4-2 and ZmEPFL5, such that ZmEPFL4-2 is now correctly grouped with ZmEPFL4-1. The updated phylogenetic tree is shown below:

Supplementary Fig. 9. Phylogenetic analysis of the *EPF/EPFL* gene family in maize (orange), rice (blue), and Arabidopsis (black). The six genes selected for CRISPR-mediated editing are highlighted in bold orange and underlined. The right panel represents the relative expression patterns of maize EPF/EPFL family genes in the SAM and IM, based on RPM and FPKM values from publicly available RNA-seq datasets in NCBI (SRP101301 and PRJNA911902).

8. Line 181; Authors state, “we selected six EPFL genes that were highly expressed in the SAM and IM, based on the available RNA seq”. Fig. 8. Based on that Figure ZmEPFL9-1 is not the most expressed gene. Either ZmEPFL4-2, ZmEPFL8, or ZmEPFL2-2 would be a better choice. Why was ZmEPFL9-1 selected?

Thank you for your question. We agree that the expression of *ZmEPFL9-1* in the SAM and IM is lower than that of *ZmEPFL4-2*, *ZmEPFL8*, or *ZmEPFL2-2*. At the time of gene selection, we had some preliminary evidence that *ZmEPFL9* can bind to *ZmER1*, which motivated us to include *ZmEPFL9* as a candidate despite its relatively low expression level. Due to technical limitations of the CRISPR array, we were not able to simultaneously target *ZmEPFL4-2*, *ZmEPFL8* and *ZmEPFL2-2* in this study. Future knockout studies of these genes will be important to fully elucidate their functions. To be more accurate, we have also revised the text from “we selected six *EPFL* genes that were highly expressed in the SAM and IM” to “we selected six *EPFL* genes that were relatively highly expressed in the SAM and IM” in the revised manuscript.

9. The top labels of Fig.5a are incorrect. ZmER2-KD-AD and ZmERL-KD-AD are labeled as ZmER1-KD-AD.

Thank you for catching this mistake. And the mislabeling was corrected in the revised Fig.5a.

10. Describe the logic of using ZmFEA3 and ZmBIP1 as controls for a split-luciferase assay in the methods. Why these two genes?

Thank you for your question. As ZmERs and ZmCRN proteins both membrane-localized proteins, we choose *ZmFEA3* and *ZmBIP1*, which are also membrane localized, to serve as the control. The logic was now added in the methods as follows:

“Two membrane localized proteins, *ZmFEA3* and *ZmBIP1*, were choose as the control in the split-luciferase assay. The CDS of *ZmFEA3* and *ZmBIP1* were cloned into pCAM1300-NLuc and pCAM1300-CLuc respectively and transformed into agrobacterium GV3101.”

11. Use of gene symbols and full names in Supplemental data #2 and #3 (columns I and J) is random. For example, *ZmWUS1*, which is actively discussed in the results, is not labeled, while some other genes that are never discussed are labeled. Why? I recommend labeling as many genes as possible.

Thank you for the suggestion. We have now carefully annotated the gene information by adding gene symbols and full names in columns I and J, respectively. An additional 442 DEG genes in *Zmer1;Zmer2* and 489 DEG genes in *Zmer1;Zmerl* are newly annotated with both gene

symbols and full names. *ZmWUS1* and other discussed genes were all labeled now in the revised Supplemental data #2 and #3.

12. Fig. 6. Does functional enrichment analysis of RNA-seq data identify any biological themes or functions?

Thank you for the valuable suggestion. As recommended, we performed Gene Ontology (GO) enrichment analysis of the differentially expressed genes in *Zmer* mutants. The analysis revealed that DEG genes were enriched in multiple biological processes, including those related to hormone regulation and response, as well as developmental regulation (eg. response to hormone, regulation of hormone levels, cell fate specification, and meristem specification). These data have been included in the newly added **Supplementary Fig.11** and incorporated into the revised manuscript as follows:

“...Gene Ontology (GO) enrichment analysis revealed that DEGs in both mutants were enriched for biological processes related to hormone regulation and response, as well as developmental regulation (Supplementary Fig.11). Consistent with this, a set of genes involved in regulation of development, meristem activity and hormone biosynthesis and signaling were significantly differentially expressed in both mutants (Fig. 6b-c)...”

Supplementary Fig.11. Gene Ontology (GO) enrichment analysis of differentially expressed genes in *Zmer1;Zmer2* and *Zmer1;Zmer1*. **a** In *Zmer1;Zmer2*, upregulated and downregulated genes were significantly enriched in multiple biological themes, including response to hormone, regulation of hormone levels and inflorescence development. **b** In *Zmer1;Zmer1*, upregulated genes were significantly enriched in biological themes such as response to hormone and regulation of

cell fate specification and no significant enrichment was detected among down-regulated genes. Bar plots show the significance of enriched GO biological process (BP) terms ($-\log_{10}(\text{P value})$), with circle sizes indicating the number of associated genes.

Reviewer #2 (Remarks to the Author):

Remarks to the author

This is a very thorough characterization of the function of the ERECTA pathway in maize including the identification of three receptors and five ligands and role in vegetative and reproductive development. My only concern was with the writing of the discussion which basically repeated the results. Please compare and contrast the results to the previously published work in Arabidopsis and rice.

Thank you for the comments and suggestions!

We have now added further discussion comparing our results with previously published works. The newly added discussion are as follows:

“...Our genetic analyses indicate that ZmER1 plays the primary role in regulating meristem development and plant architecture, with ZmER2 and ZmERL providing partially redundant functions. This genetic relationship aligns with that observed in arabidopsis, where *er* mutants display compact shoots and inflorescences, and *er11* and *er12* act redundantly to enhance the *er* phenotypes^{25,26}. However, in maize, ZmER2 appears to contribute more strongly than ZmERL to inflorescence meristem (IM) size and ear architecture, as *Zmer1;Zmer2* double mutants exhibit more severe fasciation than *Zmer1;Zmer1* plants, consistent with the closer phylogenetic relationship between ZmER1 and ZmER2 (Fig. 2). This differs slightly from arabidopsis, where *er11* and *er12* enhance the *er* phenotype in partially overlapping ways, but no clear functional hierarchy between them has been established²¹. Notably, ZmER1 appears to have a more pronounced effect on meristem size than its arabidopsis counterpart. *Zmer1* single mutants already display significantly enlarged SAMs and IMs, and meristem size is further increased in *Zmer1* double and triple mutant combinations. By contrast, arabidopsis single and double *er* family mutants show little to no change in SAM size, and only the *er er11 er12* triple mutant exhibits a dramatically broadened SAM²⁵. These findings indicate that the maize ERECTA family likely exerts a stronger quantitative influence on meristem maintenance than in arabidopsis.

Importantly, disruption of *ZmER* genes directly increase kernel row number, linking ERECTA signaling to yield-related phenotypes. A similar function has been reported in rice, where *oser1* mutants exhibit increased spikelet number per panicle³¹. These findings together identify ERECTA as a conserved target for breeding or genome editing to improve inflorescence trait and yield potential. In addition, *Zmer1* and *Zmerl* single mutant, as well as all three double mutants, have an increased stomatal index, indicating that *ZmER* genes contribute to stomatal development (Supplementary Fig.15). This is consistent with the well-characterized function of the ERECTA family in arabidopsis as negative regulators of stomatal development^{58,59}.

In arabidopsis, four ER ligands EPFL1, EPFL2, EPFL4 and EPFL6 act redundantly to constrain meristem size and promote leaf initiation. The *epfl1,2,4,6* quadruple mutants have a compact shoot and inflorescence architecture similar to that of the *er* mutant¹⁸. In rice, the four OsER ligands OsEPFL6, OsEPFL7, OsEPFL8, OsEPFL9 act synergistically to control rice panicle morphogenesis. Single *Osepf16*, *Osepf18* and *Osepf19* mutant already have enhanced spikelet number, and double mutants show an even greater increase^{32,60}. These rice ligands also affect plant architecture in single and double mutant combinations, suggesting overlapping roles in both vegetative and reproductive development. Similarly, our results indicate that maize EPFL signaling also contributes to inflorescence meristem regulation. We generated CRISPR-cas9 knockout alleles in five *ZmEPFL* genes, including *ZmEPFL1-1*, *ZmEPFL1-2*, *ZmEPFL2-1*, *ZmEPFL4-2* and *ZmEPFL9-1*, which are relatively highly expressed in IM and SAM. Among the mutant combinations identified, double and triple mutants that include *Zmepfl1-1* display enlarged, fasciated IMs and the quintuple mutants showed the most pronounced IM phenotype, while displaying no obvious changes in overall plant architecture. Together, these findings support a broadly conserved EPFL-ER signaling module that tunes meristem size and inflorescence architecture across species. Further analysis of additional allele combinations will be required to clarify the precise function of each *ZmEPFL* gene and the extent of functional redundancy among them...”

“...In arabidopsis, genetic evidence demonstrates that ER family and CLV3 signaling synergistically regulate meristem activities, with the *clv3 er1 erl1 erl2* quadruple mutant displaying the most dramatic meristem phenotype^{27,28,29,61}. The ERECTA family signaling acts via EPFL ligands to laterally restrict *WUS* expression, complementing the CLV3-mediated central repression^{28,29}. In *er1 erl1 erl2*, *WUS* expression is upregulated, and genetic analyses further

revealed that WUS is epistatic to ERECTA pathway, as *wus er1 er11 er12* exhibit a *wus*-like meristem and shoot phenotype²⁹. In maize, we similarly found that *ZmER* genes negatively regulate *ZmWUS1* expression, with *ZmWUS1* being significantly upregulated in the IMs of the *Zmer1* single mutant as well as in *Zmer1;Zmer2* and *Zmer1;Zmer1* double mutants. Mutation of *Zmwus1* significantly, though not completely, suppressed the enlarged IM of *Zmer1*. However, *Zmwus1* cannot rescue the compact plant architecture of *Zmer1*. These findings suggest that the role of ZmER1 in IM regulation rely on ZmWUS1, whereas its function in vegetative architecture may involve additional or distinct components, highlighting both conserved and divergent aspects of ERECTA-WUS interactions across species.

In maize, we found that ZmER1 genetically interacts with the key CLV component ZmCRN. The *Zmer1;Zmcrn* double mutant displayed a further increase in IM size compared with either single mutant, accompanied by stronger upregulation of *WUS* expression (Supplementary Fig 16). These findings indicate that ZmERs and CLV pathways genetically interact in maize, reminiscent of the ER-CLV interactions reported in arabidopsis^{29,62}. Interestingly, we also detected a physical interaction between ZmER1 and ZmCRN, as revealed by yeast two-hybrid, split-luciferase and co-immunoprecipitation assays, suggesting the possibility of direct crosstalk between ER and CLV pathway. Given that ZmCRN is known to function together with the CLV receptor FEA2, its interaction with ZmER1 suggests that ZmCRN may play dual roles in both CLV and ER signaling. Notably, despite the physical interaction with ZmER1, the *Zmcrn* mutants did not exhibit the vegetative architecture phenotypes seen in *Zmer1*, suggesting a complex relationship and non-identical functionality outputs of these two factors (Fig. 5). Together, our data support an interconnection between ER and CLV-WUS pathways and suggest a working model in which ZmER1 perceives EPFL peptides and interacts with ZmCRN to regulate in IM development, possibly in part by constrain *ZmWUS1* expression (Supplementary Fig.17)....”

Information that needs to be added.

Figure 1 – it is difficult to distinguish the expression of ZmER1 from the Turbo-YFP pictures. Can you add an RNA in situ hybridization so that the expression can be compared to the other two ER genes.

Thank you for your suggestion. We now added the *in situ* hybridization data of *ZmER1* in the shoot apical meristem (SAM) which also showed that *ZmER1* is expressed throughout the SAM, with relatively higher expression levels in the leaf primordia.

This manuscript was revised accordingly as follows:

" ...In addition, *in situ* hybridization experiments also showed that *ZmER1*, *ZmER2* and *ZmERL* were all expressed in the SAM, with stronger signals in leaf primordia (Fig. 1g). "

g

Fig. 1g. *In situ* hybridization analysis showed *ZmER1*, *ZmER2*, and *ZmERL* were expressed in the SAM, with stronger signals in the leaf primordia. Scale bar, 100 μ m.

Since *ERECTA* is well known to affect stomatal development in *Arabidopsis*, where there are any stomatal defects in the *Zmer* mutants?

Thank you for the question. As suggested, we examined the stomatal epidermal patterning in fully expanded leaves of the single and double mutants. Statistical analysis showed that, except for *Zmer2* single mutant, which was comparable to WT, the *Zmer1* and *Zmer1* single as well as all three double mutants, exhibited a significant increase in stomatal index. Notably, the *Zmer1;Zmer1* double mutant showed the most pronounced increase (The newly added **Supplementary Fig. 15**, shown below). Unfortunately, we did not recover triple mutants in this round of analysis due to low segregation, to assess whether they would display the strongest stomatal phenotype. Nevertheless, these result support a role for *ZmER* genes in stomatal development.

a

Supplementary Fig. 15. *ZmER* genes affect stomatal development. **a** Representative image of abaxial epidermal showing stomatal patterning in *Zmer* single and double mutant. **b** Quantification of the stomatal index for the indicated genotypes. Except for the *Zmer2* single mutant, all other single and double mutants displayed a significantly higher stomatal index, with the *Zmer1;Zmer1* double mutant showing the largest increase. Bars represent mean \pm s.d. (n>10). Different letters indicate significantly distinct groups as determined by one-way ANOVA. Scale bar, 100 μ m.

These results have been added in the revised manuscript as follows:

“...In addition, *Zmer1* and *Zmer1* single mutant, as well as all three double mutants, have an increased stomatal index, indicating that *ZmER* genes contribute to stomatal development (Supplementary Fig.15). This is consistent with the well-characterized function of the ERECTA family in arabidopsis as negative regulators of stomatal development^{58,59}.....”

Figure 7 – please explain how kernel row number was measured. This information was not in the materials and methods.

Thanks for pointing this out, as you suggested, we added the information in the methods as follows:

“Kernel row number (KRN) was measured on mature maize ears. Husk was removed, and the number of kernel rows was counted around the circumference at the midpoint of each ear.”

Suppl Fig 4 – what was the internode length of the mutants? Where the short internodes due to smaller cells?

Thank you for the question. We quantified internode length in WT, *Zmer1*, and *Zmer1;Zmer2* double mutants. Compared with WT, *Zmer1* plants exhibited shorter internodes, and the internode length was further reduced in the *Zmer1;Zmer2* double mutant (newly added **Supplementary Fig. 5a**, shown below), consistent with the reduced plant height of *Zmer1* and the more compact architecture of *Zmer1;Zmer2*. Analysis of longitudinal sections of the internode further revealed that the cell size in *Zmer1* was significantly smaller than WT, and was further reduced in the *Zmer1;Zmer2* double mutant (newly added **Supplementary Fig. 5b-c**, shown below.).

This information has been incorporated into the revised manuscript as follows:

“Internode length and cell size were further reduced in *Zmer1;Zmer2* compared with *Zmer1*, consistent with the greater reduction in plant height (Supplementary Fig. 5).”

Supplementary Fig. 5. The internode length and cell size were reduced in *Zmer1* and *Zmer1;Zmer2*. **a** Measurement of internode length in mature WT, *Zmer1*, and *Zmer1;Zmer2* plants. Internodes were counted from the base, with the lowest internode designated as the first. Different internodes were indicated with different colors. Error bars represent standard deviation, $n \geq 7$. **b** Longitudinal sections of the third internode below the ear at the 12-leaf stage in WT, *Zmer1*, and *Zmer1;Zmer2* plants. Scale bars: 50 μm . **c** Quantification of cell size in WT, *Zmer1*, and *Zmer1;Zmer2*. Error bars represent mean \pm s.d., * $P < 0.05$, ** $P < 0.01$.

Suppl Fig 8 – add at least the Arabidopsis and rice genes to the ZmEPFL phylogeny. The relationships are impossible to determine otherwise.

Thank you for the suggestion. We have now incorporated the EPF/EPFL homologs from arabidopsis and rice to generate a more comprehensive phylogenetic tree, which clarifies how the *ZmEPF/EPFL* gene names correspond to their counterparts in arabidopsis and rice. The updated phylogenetic tree is shown below:

Supplementary Fig. 9. Phylogenetic analysis of the *EPF/EPFL* gene family in maize (orange), rice (blue), and Arabidopsis (black). The six genes selected for CRISPR-mediated editing are highlighted in bold orange and underlined. The right panel represents the relative expression patterns of maize EPF/EPFL family genes in the SAM and IM, based on RPM and FPKM values from publicly available RNA-seq datasets in NCBI (SRP101301 and PRJNA911902).

Minor concerns

Figure 3 – there are two typos in parts d and e. “quintuple”,

Thank you for catching the typos. We corrected them in the figures and proofread the manuscript.

Where off target effects checked in the EPF and EPFL CRISPR lines?

Thank you for this question. We did not perform whole-genome sequencing to assess potential off-target edits across the genome. However, we conducted two rounds of backcrossing to eliminate background mutations and the six *EPFL* genes were amplified and sequenced by Sanger sequencing to validate gene edits in each generation. In the revised manuscript, we now clarify the backcrossing procedure as follows:

“...Lines carrying mutations in different *ZmEPFL* genes were backcrossed twice with the KN5585 inbred line to segregate away off-target mutations and remove the Cas9 transgene to avoid further edits.”

Figure 6 – since *TS1* and *SK1* were differentially expressed – where there sex determination effects in the *ZmER* double mutants? Do you have an explanation for why these genes are increased in expression?

Thank you for the question. Actually, we did not observe any obvious sex-determination phenotypes in the *Zmer* double mutants (i.e., no obvious feminized tassel or ears without pistil). We do not have a definitive explanation for the increased expression of *TS1* and *SK1*. Our GO enrichment indicates broad up-regulation of hormone-related processes in the *Zmer* mutant. Considering that *TS1* (a lipoxygenase) and *SK1* (a UDP-glycosyltransferase) act on jasmonate metabolism and signaling, with *TS1* promotes JA production while *SK1* reduces levels of active jasmonates, we speculated that their upregulation in *Zmer* mutants likely reflect compensatory feedback in the JA pathway or an indirect effect of altered hormone/signaling homeostasis. *TS1* promotes JA production while *SK1* counteracts JA activity. Their concurrent up-regulation may have internal buffering that keep effective JA to keep normal pistil elimination. In addition, sex determination in maize is highly stage and organ-specific. Our RNA seq analysis were carried out in florescence meristem, not in the floral meristem or floral organ, the elevated *TS1* and *SK1* may occur outside the critical domains/times that control pistil fate. That’s possible explain why we didn’t see a sex determination effect. We also add a bit discussion in the revised manuscript as follows:

“...In addition, we also detected concurrent upregulation of two jasmonate-pathway genes, *TS1*, encoding a lipoxygenase involved in JA biogenesis, and *SKI*, which encode a UDP glycosyltransferase that attenuates active jasmonates, which implies potential modulation of JA signaling in these mutants^{65,66}. Nevertheless, no obvious sex determination phenotypes were observed, suggesting the JA homeostasis maybe buffered in the developing florets...”

Is ref1 the correct ref? Include original references to *td1*, *fea2* etc?

Thank you for the suggestion. We agree that *ref1* may not be the most appropriate reference here. As recommended, we have removed it and added the original references related to *TD1* and *FEA2*, as follows:

Bommert, P. *et al.* thick tassel dwarf1 encodes a putative maize ortholog of the Arabidopsis CLAVATA1 leucine-rich repeat receptor-like kinase. *Development*. **132**, 1235–1245 (2005).

Bommert, P., Nagasawa, N. S. & Jackson, D. Quantitative variation in maize kernel row number is controlled by the FASCIATED EAR2 locus. *Nat. Genet.* **45**, 334–337 (2013).

Taguchi-Shiobara, F., Yuan, Z., Hake, S. & Jackson, D. The *fasciated ear2* gene encodes a leucine-rich repeat receptor-like protein that regulates shoot meristem proliferation in maize. *Genes Dev.* **15**, 2755–2766 (2001).

Suppl Fig 1 ER phylogeny – since the rice genes are published please add their names beside the gene ID's as was done for Arabidopsis and Maize

Thank you for the suggestion. We have added the rice names beside the gene IDs in the updated **Supplementary Fig.1a**, as shown below:

Supplementary Fig.1a. Phylogenetic analysis revealed that the arabidopsis *ER* has two orthologs genes in maize, *ZmER1* and *ZmER2*, while *AtERL1* and *AtERL2* correspond to a single ortholog, *ZmERL* in maize. The rice *OsER1*, *OsER2* and *OsERL* are also included in the tree.

Suppl Fig 5 and Fig 7 RPM of maize ER genes – add references to where this data came from

Thank you for the suggestion, in the revised manuscript we have added the references for the source of maize ER gene RPM data presented in these figures. The references are listed as follows:

Knauer, S. *et al.* A high-resolution gene expression atlas links dedicated meristem genes to key architectural traits. *Genome Res.* **29**, 1962–1973 (2019).

Sun, Y. *et al.* Progressive meristem and single-cell transcriptomes reveal the regulatory mechanisms underlying maize inflorescence development and sex differentiation. *Mol. Plant* **17**, 1019–1037 (2024).

Suppl Fig 6 –The WT control here included the transgene. Since the wt without transgene was not shown, where there any additional phenotypes from adding a second copy of ZmER?

Thank you for the question, we did not observe any obvious changes in overall plant and ear architecture. For clarity, we have now included the WT without transgene in the newly updated **Supplementary Fig. 7b-c**.

Supplementary Fig.7b-c. The *ZmER1* transgene successfully rescued the dwarf and fasciated ear phenotypes of the *Zmer1* mutant. Scale bars, 20 cm (b) and 10 cm (c).

Reviewer #3 (Remarks to the Author):

This study from the groups of Drs. Jackson and Xu investigates the functions of the conserved ERECTA (ER) family members and their ligands, EPIDERMAL PATTERNING FACTOR-Like (EPFL) peptides, in regulating shoot meristem size, plant architecture, and ear development in maize. By integrating CRISPR-mediated gene editing, molecular genetic analysis, biochemical assays, and transcriptomic studies, the authors show that a member of the ER family, directly interacts with ZmCRN to restrict ZmWUS1 expression, thereby controlling meristem development. The majority of the datasets presented are solid and convincing, and the ZmER1-mediated genetic pathways proposed in this work offer valuable insights into the conservation and divergence of key regulatory components and their family members in controlling shoot meristem development in flowering plants.

Thank you for the comments!

The reviewer only has a few comments and suggestions that may help strengthen the manuscript:

1. Proposed Genetic Pathway of ZmER1–ZmWUS1. The authors demonstrated that ZmWUS1 is upregulated in *Zmer1;Zmer2* and *Zmer1;Zmerl* mutants based on both RNA-seq and qPCR data, and further showed that the enlarged meristem phenotype of *Zmer1* is dependent on ZmWUS1. These findings are interesting, but a few points need to be addressed: 1) In Fig. 6f, the authors showed that *Zmwus1* suppressed the enlarged inflorescence meristem (IM) phenotype of *Zmer1*. To make this result more convincing, please indicate how many independent biological replicates were included in the analysis and provide quantitative measurements of meristem size across the four genotypes. 2). The plant growth images in Fig. 6e suggest that *Zmwus1* moderately rescues the dwarf phenotype of the *Zmer1* mutant. Is this because plant architecture and meristem size can be uncoupled, and ZmER1 specifically regulates meristem development through ZmWUS1? A clearer interpretation of this observation would be helpful. Additionally, is ZmWUS1 upregulation specific to the meristem, or is it also elevated in non-meristematic tissues? RNA in situ hybridization of ZmWUS1 in *Zmer1* single or double mutants would help address this question. 3) Following up on this point: are there any misregulated genes in *Zmer1* mutants that are dependent on functional ZmWUS1?

Thank you for the suggestions. We measured the sizes of inflorescences meristems (IMs) in the wild type (WT), *Zmwus1* and *Zmer1* single mutants, as well as *Zmer1;Zmwus1* double mutant, and performed statistical analyses using independent biological replicates, as shown in the newly added Fig. 6i. The results indicate that the *Zmer1* IM is significantly larger than that of the WT, and this enlargement is significantly suppressed by *Zmwus1*.

Fig.6i. Quantification of IM size of WT, *Zmer1*, *Zmwus1* and *Zmer1;Zmwus1* shows that *Zmer1* significantly suppresses the enlarged IM of *Zmer1*. Although IM size in the double mutant remained bigger than the WT, the reduction was substantial. n values are shown in the bars. Data were presented as the means \pm s.d., *** $P < 0.001$, * $P < 0.05$, from an unpaired two-tailed Student's *t*-test.

In the original manuscript, we have stated that the compact plant architecture of *Zmer1* was not obviously suppressed by *Zmwus1*. However, the plant image in the original Figure 6e may have been misleading, giving the impression that *Zmwus1* moderately rescues the dwarf phenotype of the *Zmer1* mutant. We apologies for this oversight. We have now replaced the images with more

representative examples, which clearly show that *Zmer1; Zmwus1* double mutant does not differ from *Zmer1* single mutant (the updated **Figure 6e**, shown below). In addition, we have included statistical analyses of plant height and ear height (the newly added **Figure 6f and 6g**, shown below) in the revised manuscript, which confirm that there are no significant differences between *Zmer1* single mutant and *Zmer1;Zmwus1* double mutant.

The updated **Fig. 6e** and newly added **Fig.6f-g** are shown below:

Fig.6e. *Zmer1;Zmwus1* double mutants had similar height as *Zmer1* single mutants. **Fig.6f-g** Statistical analyses revealed that there were no significant differences in plant height or ear height between the *Zmer1;Zmwus1* double mutant and the *Zmer1* single mutant. Data were presented as the means \pm s.d., sample size (n) are indicated in the bars, **** $P < 0.0001$, from an unpaired two-tailed Student's *t*-test.

Our results demonstrate that while mutation in *Zmwus1* can significantly suppress the enlarged meristem of *Zmer1*, it does not alleviate the compact plant architecture. We also agree with the reviewer that these results indicated that the function of ZmER1 specifically in inflorescence meristem development depends on ZmWUS1. Furthermore, our findings suggest that IM size can be uncoupled from the plant architecture, which was also implicated in previous studies showing that variations in *Kernel Row Number 2 (KRN2)* and *KRN4* lead to enlarged IM while maintaining normal plant architecture (Chen, W et al. 2022, *Science*; Liu, L et al. 2015, *PLoS Genet.*).

As suggested, we have clarified this interpretation in the revised manuscript as follows:

“...Interestingly, the compact plant architecture of *Zmer1* was not suppressed by *Zmwus1*, as the *Zmer1;Zmwus1* double mutant showed similar plant and ear height to the *Zmer1* single mutant (Fig.6e-g). In contrast, the *Zmwus1* mutation significantly suppressed the enlarged IM phenotype of *Zmer1*, with the IM size of *Zmer1; Zmwus1* double mutant being significantly smaller than that

of *Zmer1* (Fig.6h-i). These findings suggest that *ZmWUS1* is required for the enlarged meristem phenotype of *Zmer1*, whereas the compact plant architecture of *Zmer1* appears largely *ZmWUS1*-independent. Moreover, these results also suggest that IM development can be uncoupled from the plant architecture, consistent with previous studies showing that variations in *Kernel Row Number 2* (*KRN2*) and *KRN4* lead to enlarged IMs while maintaining normal plant architecture^{48,49}...”

To investigate whether the *ZmWUS1* upregulation is specific to the meristem, or it is also elevated in non-meristematic tissues. We have carried out *in situ* hybridization as reviewer suggested. However, we were unable to detect clear *ZmWUS1* signals in our system. Detecting *ZmWUS1* expression in maize by *in situ* hybridization is very technically challenging, as its expression is very low and restricted to only a few cells in the meristem. The low expression of *ZmWUS1* is also reflected by the very low RPKM value (~0.8) in our RNA seq data. As an alternative, we have conducted a real-time quantitative PCR assay, a more sensitive and quantitative approach for measuring gene expression. Our results showed that *ZmWUS1* upregulation was only detected in the IM region but not in non-meristematic tissues such as the vascular region within the ear primordia, young husk leaf or internodes.

Supplementary Fig. 12. Relative expression levels of *ZmWUS1* in WT and *Zmer1;Zmerl* plants in different tissues. In the inflorescence meristem (IM), *ZmWUS1* expression was significantly higher in *Zmer1;Zmerl* compared with WT (** $P < 0.01$, two-tailed Student's *t*-test). In contrast, expression was barely detectable in vascular tissues in ear primordia, young husk leaf, and internodes, with no significant difference between the two genotypes. Data are presented as mean \pm s.d.

From the RNA-seq analysis, two *CLE* genes, *ZmFCP1* and *ZmCLE21*, which may be dependent on functional *ZmWUS1*, were also found to be upregulated. This information has now been clearly stated in the revised manuscript as follows:

“...Besides, two *CLE* genes, *ZmFCP1* and *ZmCLE21*, which are putatively *ZmWUS1* responsive, were also upregulated...”

2. Characterization of weak *Zmer1* alleles

The reviewer appreciates the authors' efforts in identifying and characterizing weak *Zmer1* alleles as potential resources for yield improvement. However, there appears to be a discrepancy between the two weak alleles analyzed. The *Zmer1A369T* allele shows a statistically significant increase in kernel row number (KRN) compared to WT, whereas the difference between WT and *Zmer1D295N* is not statistically supported (Fig. 7g). Additionally, several other traits appear unchanged in both alleles compared to WT (Fig. 7h-j). Given these observations, the interpretation regarding the yield-enhancing potential of *Zmer1* weak alleles should be made more cautiously.

Thank you for the suggestion. Our data showed the *Zmer1^{A369T}* allele significantly increased kernel row number (KRN), while *Zmer1^{D295N}* showed a similar trend, but the increase was not statistically significant (Fig. 7g). This difference in KRN improvement was not unexpected, as different alleles often have distinct effect. Similar observations have been reported for weak alleles of *fea2* and *fea3*, where KRN varied depending on the specific alleles (Bommert, P *et al.*, 2013, Nat Genet; Je, B *et al.*, 2016, Nat Genet). Previous studies have also shown that traits such as the kernel number per row, ear length, or hundred-kernel weight, can be reduced in strong fasciation mutant with extreme KRN increases, whereas these traits are often unaffected in weak mutants (Bommert, P *et al.*, 2013, Nat Genet; Je, B *et al.*, 2016, Nat Genet; Liu, L *et al.*, 2021, Nat Plants). Therefore, it is reasonable that these traits were not affected in either of the *Zmer1* alleles, as both behave as weak alleles (Fig. 7h-j).

We agree with reviewer that our interpretation should be more cautious. To be more precise, in the Result section, we have avoided emphasizing the improvement of *Zmer1^{D295N}* on ear architecture, since its increase in KRN was not statistically significant. The manuscript has been revised accordingly as follows:

“These findings suggest that *Zmer1^{D295N}* and *Zmer1^{A369T}* function as weak alleles, with both alleles significantly improving leaf angle, and *Zmer1^{A369T}* also enhancing ear architecture.”

In addition, we carefully revised the Discussion section to more accurately reflect the yield-enhancing potential of weak *Zmer1* alleles as follows:

“...In this study, we showed that two weak alleles of *Zmer1* exhibited variable beneficial effects on plant and ear architecture, highlighting the potential to improve yield under high-density planting condition by genetically manipulating the *ZmER1* locus. Nevertheless, their effect should be further evaluated through large-scale yield tests with commercial cultivars, diverse plant conditions, and multiple environments to better assess the impact of these genetic variations on yield traits⁵⁷.”

3. Proposed ZmER1–ZmCRN Interaction

The authors present compelling evidence for both physical and genetic interactions between ZmER1 and ZmCRN, providing new insight into crosstalk between the ERECTA and CLAVATA pathways in meristem regulation. However, a key piece of evidence is missing to fully support the proposed working model (Supplementary Fig. 11): What is the functional output of the ZmER1–ZmCRN interaction? Specifically, is ZmWUS1 expression further upregulated in the *Zmer1*; *Zmcrn* double mutant compared to WT or either single mutant? This analysis would help solidify the proposed model.

As suggested, we have assessed the *ZmWUS1* expression in the IM of *Zmer1* and *Zmcrn* as well as the double mutant. The result revealed the *ZmWUS1* expression was significantly increased in either single mutant and its expression was further increased in the double mutant (the newly added **Supplementary Fig. 16**). This data has been added in the revised manuscript.

“...The *Zmer1*; *Zmcrn* double mutant displayed a further increase in IM size compared with either single mutant, accompanied by stronger upregulation of *ZmWUS1* expression (Supplementary Fig. 16)....”

Supplementary Fig. 16. Relative transcriptional level of *ZmWUS1* in WT, *Zmer1*, *Zmcrn* and *Zmer1*; *Zmcrn* determined by RT-qPCR. Data are presented as mean values \pm s.d., ** $P < 0.01$, *** $P < 0.001$, by two-tailed Student's *t*-test.

Reviewer #1 (Remarks to the Author):

Previously, I had two major criticisms. The first concerned the authors' claim that they had identified ZmEPFL genes regulating inflorescence meristem (IM) development, which was not sufficiently supported by the data. The revised manuscript now includes convincing evidence linking ZmEPFL1-1 to the regulation of IM size. However, the contributions of other EPFLs remain unclear. Based on the available data, it appears that among ZmEPFL1-2, ZmEPFL2-1, ZmEPFL4-2, and ZmEPFL9-1, some peptides influence IM size in a partially redundant manner. While the authors have modified their conclusion in the text to acknowledge this point, Figure S11 still depicts all other EPFLs as regulators of WUS expression. Please revise this figure accordingly. Additionally, it would be valuable to include an image of a single *Zmepfl1-1* mutant along with quantitative measurements of its IM size. This would help determine whether the phenotypes observed in *Zmepfl1-1*, *Zmepfl1-2* and *Zmepfl1-1 Zmepfl9-1* mutants are primarily due to the *Zmepfl1-1* mutation, providing definitive evidence that ZmEPFL1-1 regulates IM size.

Thank you for the comments and suggestions. We agree with the reviewer that the model figure in Figure S17 (Figure S11 referenced by the reviewer) may not be appropriate, and as also suggested by the editor we have removed it from the revised manuscript.

At present, we have not obtained the single *Zmepfl1-1* mutant due to the low segregation ratio associated with the five loci. We speculated that enlarged IM observed in *Zmepfl1-1 Zmepfl1-2* and *Zmepfl1-1 Zmepfl9-1* mutants is primarily due to the *Zmepfl1-1* mutation, as the *Zmepfl1-2 Zmepfl9-1* double mutant doesn't have an obvious phenotype. We agree with the reviewer that ZmEPFL1-1 likely plays a prominent role in the regulation of IM size, while ZmEPFL1-2, ZmEPFL2-1, ZmEPFL4-2, and ZmEPFL9-1 may influence IM size in a partially redundant manner. We now address this in the Discussion part as follows:

“...Together, our data suggest that ZmER1 perceives EPFL peptides, primarily EPFL1-1, with potential redundancy from other EPFLs (ZmEPFL1-2, ZmEPFL2-1, ZmEPFL4-2, and ZmEPFL9-1) to regulate IM development, possibly in part by constraining *ZmWUS1* expression....”

My second criticism concerns the inclusion of ZmCRN as part of the ER signaling pathway. The authors still have not provided sufficient data to support this conclusion. In the revised version, they added one additional experiment showing that FEA2 interacts with CRN with much higher affinity compared to ZmER1. The weak interaction observed between CRN and ER1 could simply result from protein overexpression in tobacco leaves. The presented evidence is not conclusive that ZmCRN functions within the ER signaling pathway, and I

continue to strongly recommend removing this claim from the manuscript. While I agree that “it is not uncommon for one protein to participate in different protein complexes,” the current data do not convincingly demonstrate such a role for ZmCRN with ERs. All protein interaction assays (yeast two-hybrid and split-luciferase) were conducted under overexpression conditions, in which many non-specific interactions can occur. Furthermore, the reported genetic interactions are questionable, as a synergistic relationship would be expected between the FEA2 pathway and ERs. I do not understand the authors’ insistence on maintaining this unsupported conclusion in an otherwise strong manuscript. Although the manuscript text was slightly revised to temper the conclusion, the changes are not sufficient, and the model in Figure S11 still depicts ZmCRN.

All my remaining criticisms and suggestions have been addressed satisfactorily.

Thank you for the positive comments. We agree with the reviewer’s concern regarding the protein interaction. As suggested, we have removed the section describing the physical interaction between ZmER1 and ZmCRN and now refer only to the genetic interaction revealed by the *Zmer1* and *Zmcrn* double mutant analysis, which is described in the Discussion section as follows:

“...In addition, we found that ZmER1 genetically interacts with the key CLV component ZmCRN. While *Zmcrn* mutants didn’t have obvious plant architecture defects, *Zmcrn* enhanced the plant architecture phenotypes of *Zmer1* in double mutants (Supplementary Fig 15a). Furthermore, the *Zmer1;Zmcrn* double mutants further increased IM size compared with either single mutant, accompanied by a stronger upregulation of *ZmWUS1* expression (Supplementary Fig 15b and 15c). These findings indicate that ZmERs and CLV pathways genetically interact in maize, reminiscent of the ER-CLV interactions reported in arabidopsis^{29,61}”

In addition, the model figure in Figure S17 (Figure S11 referenced by the reviewer) has been removed accordingly.

Reviewer #3 (Remarks to the Author):

In the revised manuscript, the authors have included new experimental data and have revised the descriptions and discussion. I appreciate the authors’ efforts and careful revisions, and I believe that the revised manuscript has been significantly improved, addressing all of my previous comments. I have no further comments or concerns.

Thank you for the positive comments!